

# Zero temperature momentum distribution of an impurity in a polaron state of one-dimensional Fermi and Tonks-Girardeau gases

**Oleksandr Gamayun[1⋆], Oleg Lychkovskiy[2,3†] and Mikhail B. Zvonarev[4,5‡]**

**1** Institute for Theoretical Physics, University of Amsterdam, Science Park 904, Postbus 94485, 1090 GL Amsterdam, The Netherlands
**2** Skolkovo Institute of Science and Technology, Skolkovo Innovation Center 3, 143026 Moscow, Russia
**3** Steklov Mathematical Institute of Russian Academy of Sciences, 8 Gubkina St., Moscow 119991, Russia
**4** Université Paris-Saclay, CNRS, LPTMS, 91405, Orsay, France
**5** St. Petersburg Department of V.A. Steklov Mathematical Institute of Russian Academy of Sciences, Fontanka 27, St. Petersburg, 191023, Russia

⋆ o.gamayun@uva.nl, † lychkovskiy@gmail.com, ‡ mikhail.zvonarev@gmail.com

## Abstract

We investigate the momentum distribution function of a single distinguishable impurity particle which formed a polaron state in a gas of either free fermions or Tonks-Girardeau bosons in one spatial dimension. We obtain a Fredholm determinant representation of the distribution function for the Bethe ansatz solvable model of an impurity-gas $\delta$-function interaction potential at zero temperature, in both repulsive and attractive regimes. We deduce from this representation the fourth power decay at a large momentum, and a weakly divergent (quasi-condensate) peak at a finite momentum. We also demonstrate that the momentum distribution function in the limiting case of infinitely strong interaction can be expressed through a correlation function of the one-dimensional impenetrable anyons.

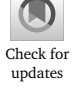

# 1  Introduction

Non-interacting Bose and Fermi systems have markedly different momentum distribution functions at low temperature. Bosons tend towards a macroscopic occupation of the zero-momentum state, and fermions spread over the volume of the Fermi sphere. When interparticle interactions are present, the distinction becomes not at all evident. We know from exactly solvable models in one spatial dimension that some observables evolve smoothly from boson- to fermion-like behavior, as a function of the inter-particle interaction strength. An example is provided by the Lieb-Liniger model, representing a gas of bosons interacting through a $\delta$-function potential of an arbitrary strength $g$ [1, 2]. The excitation spectrum of the model in the $g \to \infty$ limit, the Tonks-Girardeau gas, is the same as the one of a free Fermi gas [3, 4]. Furthermore, any excitation in the Lieb-Liniger gas is parametrized by a set of distinct integers, same way as for a free Fermi gas, giving rise to the notion of the Pauli principle for one-dimensional interacting bosons [5]. This is consistent with the fact that the low-energy and momentum excitations of the interacting gapless one-dimensional Bose and Fermi systems can be interpreted as collective boson modes of a unique effective field theory, the procedure

called the bosonization [6, 7]. Despite of these similarities, the momentum distribution functions of the Tonks-Girardeau and free Fermi gases are radically different, which is seen from the exact [8, 9], as well as asymptotic formulas [10, 11].

How do interactions shape the momentum distribution function of a single distinguishable mobile particle, an impurity, interacting with a one-dimensional system? It has been demonstrated in Ref. [12] that the function $n(k)$, defined as the probability to find the impurity in the state having the momentum $k$, does not have a single-particle delta-peak $\delta(k)$ in one spatial dimension, for any non-zero value of the impurity-gas coupling strength. Instead, $n(k) \sim k^\nu$ in the $k \to 0$ limit. The value of $\nu$ was found only in the limit of the vanishing impurity-gas coupling strength [12]. Extending this result to an arbitrary coupling strength is a far-from-trivial problem. This is because the many-body spectrum of the whole system contains low-energy excitations with quadratic dispersion relation. The application of the bosonization technique is not straightforward for such a spectrum [13, 14]. The recently developed paradigm of the non-linear Luttinger liquids [15] could perhaps be used to find $\nu$ for an arbitrary interaction strength. However, this has yet to be done. As for finding the exact shape of $n(k)$ in the whole range of values of the momentum $k$, the Bethe ansatz solution remains the only non-perturbative analytical approach available thus far.

In the present paper we investigate the shape of the momentum distribution function $n(k, Q)$ of an impurity interacting with a free Fermi gas in one spatial dimension. The system stays in the polaron state, defined as the minimum energy state at a given total momentum $Q = P_{\mathrm{imp}} + \sum_{j=1}^{N} P_j$ (Ref. [12] is dealing with $Q = 0$ only). The impurity has the same mass as the gas particle, and interacts with the gas through a $\delta$-function potential of an arbitrary (positive or negative) strength $g$. The Hamiltonian reads

$$H = \frac{P_{\mathrm{imp}}^2}{2m} + \sum_{j=1}^{N} \frac{P_j^2}{2m} + g \sum_{j=1}^{N} \delta(x_j - x_{\mathrm{imp}}). \tag{1}$$

Here, $x_j$ ($P_j$) is the coordinate (momentum) of a gas particle, $j = 1, \ldots, N$, and $x_{\mathrm{imp}}$ ($P_{\mathrm{imp}}$) is the one of the impurity. Such a model is Bethe ansatz solvable; its eigenfunctions and spectrum have been found by McGuire [16, 17]. McGuire's solution is a special case of the Bethe ansatz solution for the Gaudin–Yang model [18–20], having a peculiarity that any eigenfunction can be written as a single determinant resembling the Slater determinant for the free Fermi gas [21–23]. Such a representation, so far not available for any other interacting Bethe ansatz solvable model, enabled the derivation of an exact analytical expression for the time-dependent two-point impurity correlation function at zero [24] and arbitrary temperature [25]. Here, we present an exact analytical expression for $n(k, Q)$ in the limit of infinite system size, $L \to \infty$, valid for an arbitrary (positive or negative) coupling strength $g$ and zero temperature. The answer is given in terms of the Fredholm determinant of a linear integral operator of integrable type (see, e.g, section XIV.1 of [5]). We use our exact analytical result (i) To obtain the large-momentum tails of $n(k, Q)$, and the root mean-square uncertainty of the average momentum of the impurity. (ii) To extract a quasi-condensate-like divergence of $n(k, Q)$ at $k = Q$. (iii) To establish the correspondence between $n(k, Q)$ in the $g \to \infty$ limit and a correlation function of the one-dimensional impenetrable anyons.

The paper is organized as follows. In section 2 we define the model under consideration. In section 3 we summarize our exact analytical results expressed in terms of the Fredholm determinants. In sections 4 through 7 we analyze various limiting cases of the formulas from section 3. Section 8 explains principal steps of the calculation used to get the Fredholm determinant representation of section 3. We conclude in section 9. The appendices are self-explanatory.

## 2 Model

Our objective is to compute the momentum distribution function of an impurity,

$$n(k,Q) = \frac{L}{2\pi} \langle \min_Q | \psi_{k\downarrow}^\dagger \psi_{k\downarrow} | \min_Q \rangle, \tag{2}$$

interacting with a free one-dimensional spinless Fermi gas at zero temperature. Here, $|\min_Q\rangle$ is a polaron state, defined as the minimum energy state of the system having the total momentum $Q$ and containing only one impurity. We discuss the properties of the polaron state later in this section. Note that our result for the function (2) is also valid for the impurity immersed into the Tonks–Girardeau gas. This can be explained using the arguments given in the end of section 2 in Ref. [25].

The Hamiltonian of the entire system is

$$H = H_\uparrow + H_{\text{imp}}, \tag{3}$$

where

$$H_\uparrow = \int_0^L dx\, \psi_\uparrow^\dagger(x) \left( -\frac{1}{2m} \frac{\partial^2}{\partial x^2} \right) \psi_\uparrow(x) \tag{4}$$

is the Hamiltonian of the free Fermi gas, $m$ is the particle mass, and

$$H_{\text{imp}} = \int_0^L dx \left[ \psi_\downarrow^\dagger(x) \left( -\frac{1}{2m} \frac{\partial^2}{\partial x^2} \right) \psi_\downarrow(x) + g \psi_\uparrow^\dagger(x) \psi_\downarrow^\dagger(x) \psi_\downarrow(x) \psi_\uparrow(x) \right]. \tag{5}$$

The creation (annihilation) operators $\psi_\sigma^\dagger$ ($\psi_\sigma$) carry the subscript $\sigma = \uparrow$ for the spinless Fermi gas, and $\sigma = \downarrow$ for the impurity. We have

$$\psi_\sigma^\dagger(x) = \frac{1}{\sqrt{L}} \sum_p e^{-ipx} \psi_{p\sigma}^\dagger, \qquad p = \frac{2\pi n}{L}, \qquad n = 0, \pm 1, \pm 2, \ldots. \tag{6}$$

The Hamiltonian (3) defines the fermionic Gaudin-Yang model [18–20], in which the number of the impurity particles,

$$N_{\text{imp}} = \int_0^L dx\, \psi_\downarrow^\dagger(x) \psi_\downarrow(x) \tag{7}$$

is arbitrary. However, the states with $N_{\text{imp}} > 1$ do not contribute to the function (2). The first-quantized form of the Hamiltonian (3) with $N_{\text{imp}} = 1$ and $N$ particles from the Fermi gas is given by Eq. (1). The Planck constant, $\hbar$, is equal to one in our units. A commonly used dimensionless form of the impurity-gas coupling strength $g$ is

$$\gamma = \frac{mg}{\rho_0}, \tag{8}$$

where

$$\rho_0 = \frac{N}{L} \tag{9}$$

is the gas density. To further simplify notations, we let

$$m = 1 \tag{10}$$

and measure all momenta in the units of the Fermi momentum,

$$k_F = \pi \rho_0 = 1. \tag{11}$$

We restore $m$ and $k_F$ in the captions to the figures.

Equation (2) can be written as

$$n(k,Q) = \frac{1}{2\pi} \int_0^L dy \, e^{iky} \varrho(y), \tag{12}$$

where

$$\varrho(y) = L \langle \min_Q | \psi_\downarrow^\dagger(y) \psi_\downarrow(0) | \min_Q \rangle \tag{13}$$

is the $Q$-dependent reduced density matrix of the impurity. The normalization condition

$$\sum_k n(k,Q) = \frac{L}{2\pi} \tag{14}$$

is equivalent to

$$\varrho(0) = 1. \tag{15}$$

For the system in a finite volume $L$, periodic boundary conditions are imposed. That $n(k,Q)$ is real implies the involution

$$\varrho(-y) = \varrho^*(y), \tag{16}$$

where the star stands for the complex conjugation. The symmetry

$$n(-k,Q) = n(k,-Q) \tag{17}$$

applied to Eq. (16) gives

$$\varrho^*(y) = \varrho(y), \qquad Q = 0. \tag{18}$$

In order to compute the function (2) we use a form-factor summation approach. We write

$$n(k,Q) = \sum_{p_1,p_2,\ldots,p_N} |\langle N | \psi_{k\downarrow} | \min_Q \rangle|^2. \tag{19}$$

Here,

$$|N\rangle = \psi_{p_1\uparrow} \cdots \psi_{p_N\uparrow} |0_\uparrow\rangle \tag{20}$$

is the free Fermi gas state containing $N$ fermions with the momenta $p_1,\ldots,p_N$. The vacuum $|0_\sigma\rangle$, $\sigma = \uparrow, \downarrow$, is the state with no particles, $\psi_{p\sigma}|0_\sigma\rangle = 0$. The sum in Eq. (19) is over the states whose momenta satisfy the constraint

$$k + \sum_{j=1}^N p_j = Q. \tag{21}$$

Periodic boundary conditions imply the quantization of the momenta

$$p_j = \frac{2\pi n_j}{L}, \qquad n_j = 0, \pm 1, \pm 2, \ldots, \qquad j = 1, \ldots, N. \tag{22}$$

The coordinate representation for $|N\rangle$ is the Slater determinant

$$|N\rangle = \frac{1}{\sqrt{L^N N!}} \det_N e^{i p_j x_l}, \qquad j,l = 1, \ldots, N. \tag{23}$$

All eigenstates of the Hamiltonian (1), $|\min_Q\rangle$ being one of them, have been found in Refs. [16, 17]. Let $|Q\rangle$ be an eigenstate having total momentum $Q$. Such a state is parametrized by the quasi-momenta $k_1, \ldots, k_{N+1}$ satisfying

$$Q = \sum_{j=1}^{N+1} k_j. \tag{24}$$

The energy of the state $|Q\rangle$ reads

$$E(Q) = \sum_{j=1}^{N+1} \frac{k_j^2}{2}. \tag{25}$$

Each $k_j$ should satisfy the equation

$$k_j = \frac{2\pi}{L}\left(n_j - \frac{\delta_j}{\pi}\right), \qquad n_j = 0, \pm 1, \pm 2, \ldots, \qquad j = 1, \ldots, N+1, \tag{26}$$

where

$$\delta_j = \frac{\pi}{2} - \arctan(\Lambda - \alpha k_j), \qquad 0 \le \delta_j < \pi. \tag{27}$$

Here,

$$\alpha = \frac{2\pi}{\gamma}, \tag{28}$$

where $\gamma$ is given by Eq. (8). Thus, one has a system of $N+1$ equations (26) for the variables $k_1, \ldots, k_{N+1}$ and $\Lambda$. These equations, called the Bethe equations, are coupled through Eq. (24). Any solution to this system has the following properties [17]: (i) $\Lambda$ is real. (ii) If $\alpha \ge 0$ all $k_j$'s are real. (iii) If $\alpha < 0$ either all $k_j$'s are real, or $k_1, \ldots, k_{N-1}$ are real, while $k_N$ and $k_{N+1}$ have a non-zero imaginary part, and $k_N = k_{N+1}^*$.

We will often use the following representation of the Bethe equations (26):

$$e^{ik_j L} = \frac{v_-(k_j)}{v_+(k_j)}, \qquad j = 1, \ldots, N+1, \tag{29}$$

where

$$v_\pm(q) = \frac{1}{\alpha} \frac{1}{q - k_\mp}, \tag{30}$$

and

$$k_\pm = \frac{\Lambda \pm i}{\alpha}. \tag{31}$$

Taking the derivative of Eq. (29) with respect to $\Lambda$ we get

$$\frac{\partial k_j}{\partial \Lambda} = \frac{2}{L} \frac{v_-(k_j) v_+(k_j)}{1 + \frac{2}{L}\alpha v_-(k_j) v_+(k_j)}, \qquad j = 1, \ldots, N+1. \tag{32}$$

The point of focus of our paper is $n(k, Q)$ in the thermodynamic limit, defined as the limit of infinite system size, $L \to \infty$, at a constant density

$$\rho_0 = \frac{N}{L} = \text{const} > 0, \qquad N, L \to \infty. \tag{33}$$

In what follows, we use $L \to \infty$ in place of $L, N \to \infty$ for simplicity of the notations. The choice of the boundary conditions should play no role for $n(k, Q)$ in the thermodynamic limit. The sum over momenta turns into the integral,

$$\frac{2\pi}{L} \sum_k \to \int_{-\infty}^{\infty} dk \qquad L \to \infty, \tag{34}$$

and the normalization condition (14) becomes

$$\int_{-\infty}^{\infty} dk\, n(k, Q) = 1. \tag{35}$$

In sections 2.1 through 2.3 we proceed with solving the system of Eqs. (24) and (26) in the thermodynamic limit for the state $|\min_Q\rangle$ entering Eq. (2).

## 2.1 Defining $|\mathrm{min}_Q\rangle$ for impurity-gas repulsion

In the case of the repulsive interaction, $\gamma \geq 0$, the $L \to \infty$ limit of Eq. (27) reads

$$\delta_j = \frac{\pi}{2} - \arctan\left(\Lambda - \alpha\frac{2\pi n_j}{L}\right), \qquad j = 1,\ldots,N+1, \qquad L \to \infty. \tag{36}$$

We adopt the convention that the distinct integers $n_j$ are enumerated in the increasing order, $n_1 < \cdots < n_{N+1}$. Equation (24) turns into the algebraic relation between $\Lambda$ and $Q$:

$$Q = Q^D + \Lambda Z + \frac{1}{\pi}[\arctan(\alpha + \Lambda) - \arctan(\alpha - \Lambda)] + \alpha\varphi, \tag{37}$$

where

$$Z = \frac{\arctan(\alpha - \Lambda) + \arctan(\alpha + \Lambda)}{\alpha\pi} \tag{38}$$

and

$$\varphi = \frac{1}{2\pi\alpha^2}\ln\frac{1 + (\alpha - \Lambda)^2}{1 + (\alpha + \Lambda)^2}. \tag{39}$$

The function $Q^D$ encompasses all $n_j$'s:

$$Q^D = \frac{2\pi}{L}\sum_{j=1}^{N+1} n_j - 1. \tag{40}$$

The energy (25) turns into

$$E(Q) = \frac{1}{2}\sum_{j=1}^{N+1}\left(\frac{2\pi}{L}n_j\right)^2 + E_{\mathrm{min}}(Q), \tag{41}$$

where

$$E_{\mathrm{min}}(Q) = \frac{1}{\pi\alpha} - \frac{1 + \alpha^2 - \Lambda^2}{2\alpha}Z + \Lambda\varphi. \tag{42}$$

Let

$$n_j = -\frac{N+1}{2} + j, \qquad j = 1,\ldots N+1. \tag{43}$$

Such a choice leads to $Q^D = 0$, and corresponds to the minimum energy state $|\mathrm{min}_Q\rangle$ for $-1 \leq Q \leq 1$. Equation (37) turns into

$$Q = \Lambda Z + \frac{1}{\pi}[\arctan(\alpha + \Lambda) - \arctan(\alpha - \Lambda)] + \alpha\varphi. \tag{44}$$

The parameter $\Lambda$ runs from $-\infty$ to $\infty$ when $Q$ runs from $-1$ to $1$. Equations (42) and (44) determine $E_{\mathrm{min}}$ as a function of $Q$ for $-1 \leq Q \leq 1$. The minimum energy state for $Q$ outside of that interval is parametrized by consecutive sets of $n_j$'s other than given by Eq. (43). The result is a smooth periodic function of $Q$, plotted in the left panel of Fig. 1. Note that

$$E_{\mathrm{min}}(1) = 0, \tag{45}$$

and

$$E_{\mathrm{min}}(0) = \frac{\alpha - (1 + \alpha^2)\arctan\alpha}{\pi\alpha^2}. \tag{46}$$

Therefore,

$$E_{\mathrm{min}}(1) - E_{\mathrm{min}}(0) \geq 0, \qquad 0 \leq \gamma \leq \infty \tag{47}$$

decreases from $1/2$ to zero when $\gamma$ increases from zero to infinity.

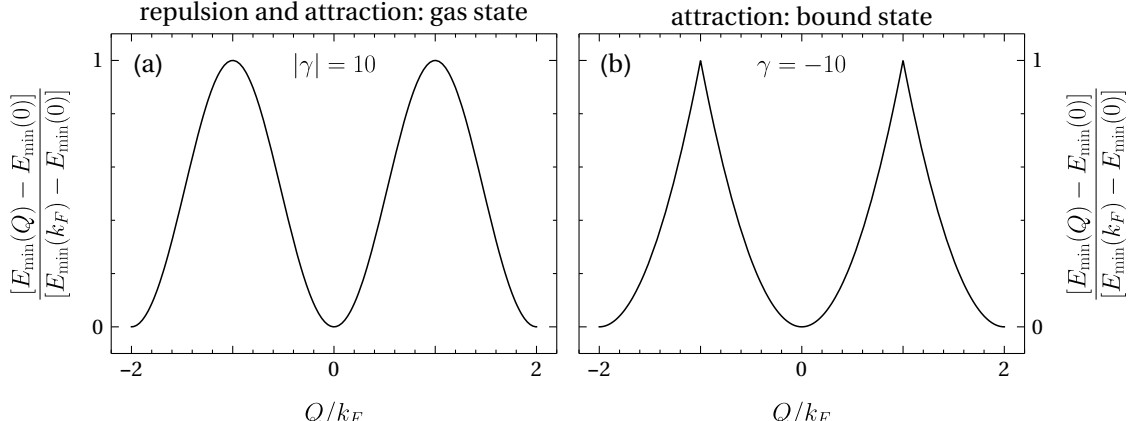

Figure 1: Shown is the normalized minimum energy $[E_{\min}(Q) - E_{\min}(0)]/[E_{\min}(k_F) - E_{\min}(0)]$ as a function of the total momentum $Q$ for the repulsive, and the attractive gas state (two identical curves, left panel), and the attractive bound state (right panel). The absolute value of the impurity-gas interaction strength is $|\gamma| = 10$. Note that $E_{\min}(Q)$ is $Q$-periodic with the period $2k_F$, it is plotted here for the two periods.

## 2.2 Defining $|\min_Q\rangle$ for impurity-gas attraction: gas state

The gas state is defined for the attractive interaction, $\gamma < 0$, as the minimum energy state for all $k_j$'s being real. Such a state has been realized experimentally for the Lieb-Liniger gas in the experiment with ultracold atoms [26]. The analysis following the steps from section 2.1 leads to Eqs. (42) and (44) in which $\gamma$ is now negative. This results in $E_{\min}(Q)$ being an odd function of $\gamma$. Therefore, the function $[E_{\min}(Q) - E_{\min}(0)]/[E_{\min}(1) - E_{\min}(0)]$ coincide with the one for the repulsive case, plotted in the left panel of Fig. 1. The function

$$E_{\min}(1) - E_{\min}(0) \leq 0, \qquad -\infty \leq \gamma \leq 0 \tag{48}$$

decreases from zero to $-1/2$ when $\gamma$ increases from minus infinity to zero. This means that the minimum energy state for a weak repulsion, $\gamma \ll 1$, does not go continuosly to the gas state for a weak attraction, $-\gamma \ll 1$. Rather, it turns into the weakly attractive bound state, discussed in section 2.3.

## 2.3 Defining $|\min_Q\rangle$ for impurity-gas attraction: bound state

The bound state is the true minimum energy state for the attractive interaction, $\gamma < 0$. That is, $k_j$'s are not required to be real, as it was for the gas state, section 2.2. As a result, the phase shifts take the form (36) for the real $k_1, \ldots, k_{N-1}$, and [17]

$$k_N = k_+ + \mathcal{O}(e^{-|g|L}), \qquad k_{N+1} = k_- + \mathcal{O}(e^{-|g|L}), \tag{49}$$

where $k_\pm$ is defined by Eq. (31). Therefore, Eq. (24) takes the form

$$Q = Q^D + \Lambda Z + \frac{1}{\pi}[\arctan(\alpha + \Lambda) - \arctan(\alpha - \Lambda)] + \alpha\varphi + k_+ + k_-, \tag{50}$$

where $Q^D$ is given by Eq. (40) with $j$ running from 1 to $N-1$. Like in the case $\gamma > 0$, we have $Q^D = 0$ for the minimum energy states in the interval $-1 \leq Q \leq 1$:

$$Q = \Lambda Z_b + \frac{1}{\pi}[\arctan(\alpha + \Lambda) - \arctan(\alpha - \Lambda)] + \alpha\varphi, \tag{51}$$

where

$$Z_b = Z + \frac{2}{\alpha}. \tag{52}$$

The function

$$E_{\min}(Q) = \frac{1}{\pi\alpha} - \frac{1+\alpha^2-\Lambda^2}{2\alpha}Z + \Lambda\varphi + \frac{k_+^2}{2} + \frac{k_-^2}{2} = 1 + \frac{1}{\pi\alpha} - \frac{1+\alpha^2-\Lambda^2}{2\alpha}Z_b + \Lambda\varphi \tag{53}$$

entering Eq. (41) is plotted in the right panel of Fig. 1, and is a periodic function of $Q$. Unlike for the repulsive and the attractive gas state, (i) $\Lambda$ runs through the finite interval, $-\Lambda_F \leq \Lambda \leq \Lambda_F$, when $Q$ runs from 1 to $-1$ in Eq. (51); (ii) $E_{\min}(Q)$ has cusps at $Q = \pm 1, \pm 2, \ldots$. One has

$$E_{\min}(0) = -\frac{1}{\alpha^2} + \frac{\alpha - (1+\alpha^2)\arctan\alpha}{\pi\alpha^2}. \tag{54}$$

The function $E_{\min}(1)$ is obtained by substituting $\Lambda_F$ into Eq. (53), and

$$E_{\min}(1) - E_{\min}(0) > 0, \qquad -\infty \leq \gamma \leq 0 \tag{55}$$

increases from $1/4$ to $1/2$ when $\gamma$ goes from minus infinity to zero.

## 3 Fredholm determinant representation in the thermodynamic limit

In this section we show the main results of our paper: exact analytic formulas for the impurity momentum distribution function $n(k, Q)$ at zero temperature and an arbitrary positive and negative impurity-gas interaction strength $g$. These formulas contain Fredholm determinants of linear integral operators. Let $V$ be an $M \times M$ matrix with the entries $V_{jl} = V(k_j, k_l)$, $I$ be the identity matrix, and

$$k_j = \frac{2(j-1)}{M-1} - 1, \qquad j = 1, \ldots, M. \tag{56}$$

Then the Fredholm determinant is

$$\det(\hat{I} + \hat{V}) = \lim_{M \to \infty} \det\left(I + \frac{2}{M-1}V\right). \tag{57}$$

The right hand side of Eq. (57) taken for a large but finite $M$ can be used to evaluate the Fredholm determinant numerically [27]. An equivalent definition,

$$\det(\hat{I} + \hat{V}) = \sum_{N=0}^{\infty} \frac{1}{N!} \int_{-1}^{1} dk_1 \cdots \int_{-1}^{1} dk_N \begin{vmatrix} V(k_1, k_1) & \ldots & V(k_1, k_N) \\ \vdots & \ddots & \vdots \\ V(k_N, k_1) & \ldots & V(k_N, k_N) \end{vmatrix}, \tag{58}$$

appears in the mathematical literature on the linear integral operators theory (see, e.g., [28], vol IV, p.24). Naturally, $\hat{V}$ can be recognized as a linear integral operator with the kernel $V(q, q')$ on the domain $[-1, 1] \times [-1, 1]$. The necessary existence and convergence conditions are fulfilled for the operators encountered in our paper.

The energy of the state $|\min_Q\rangle$ is a periodic function of $Q$, and $n(k, Q)$, defined by Eq. (2), inherits this periodicity. We rewrite Eq. (12) as

$$n(k, Q) = \frac{1}{2\pi} \int_{-\infty}^{\infty} dy\, e^{iky} \varrho(y) = \frac{1}{\pi} \text{Re}\left[\int_{0}^{\infty} dy\, e^{iky} \varrho(y)\right], \qquad L \to \infty. \tag{59}$$

In what follows, we write $\varrho(y)$ explicitly for the positive values of $y$, and use the involution (16) to get it for the negative values.

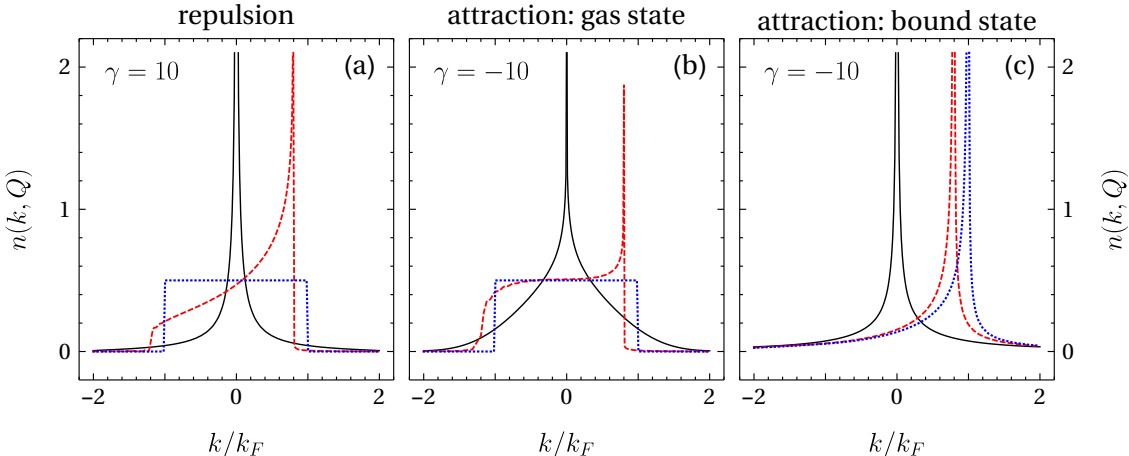

Figure 2: Impurity's momentum distribution $n(k,Q)$ is shown for different values of the total momentum: $Q = 0$ (black solid), $Q = 0.8k_F$ (red dashed), and $Q = k_F$ (blue dotted) lines, respectively. Note that $n(k,Q)$ is singular at $k = Q$. Sections 4 through 7 discuss the features revealed in this plot.

## 3.1 Impurity-gas repulsion

The Fredholm determinant representation in the case of the repulsive impurity-gas interaction, $\gamma \geq 0$, reads

$$\varrho(y) = \det(\hat{I} + \hat{K} + \hat{W}) - \det(\hat{I} + \hat{K}). \tag{60}$$

The identity operator is denoted by $\hat{I}$. The kernels of the linear integral operators $\hat{K}$ and $\hat{W}$, on the domain $[-1, 1] \times [-1, 1]$, are defined by

$$K(q, q') = \frac{e_+(q)e_-(q') - e_-(q)e_+(q')}{q - q'}, \tag{61}$$

and

$$W(q, q') = \frac{e_-(q)e_-(q')}{\pi Z}, \tag{62}$$

respectively. The kernel (61) belongs to a class of integrable kernels [5, 29]. The functions $e_\pm$ are defined as

$$e_+(q) = \frac{1}{\pi} e^{iqy/2 + i\delta(q)}, \qquad e_-(q) = e^{-iqy/2} \sin\delta(q), \tag{63}$$

and

$$Z = \frac{\delta_+ - \delta_-}{\alpha\pi}, \qquad \delta_\pm = \delta(\pm 1). \tag{64}$$

Here, the phase shift $\delta(q)$ is defined as

$$\delta(k) = \frac{\pi}{2} - \arctan(\Lambda - \alpha k), \qquad 0 \leq \delta < \pi, \tag{65}$$

and the value of $\Lambda$ can be found as a function of $Q$ by using Eq. (44). The behavior of the momentum distribution function is illustrated in Fig. 2(a).

## 3.2 Impurity-gas attraction: gas state

All formulas from the section 3.1 are valid for the gas state after letting $\gamma$ be negative. The behavior of the momentum distribution function is illustrated in Fig. 2(b).

### 3.3 Impurity-gas attraction: bound state

The presence of the bound state qualitatively affects the Fredholm determinant representation for the function $\varrho(y)$, as compared with Eq. (60):

$$\varrho(y) = e^{-i(k_+ + k_-)y} \left[ \det(\hat{I} + \hat{K}_b + \hat{W}_b) - c \det(\hat{I} + \hat{K}_b) \right]. \tag{66}$$

Here,

$$c = 1 - \frac{2e^{ik_- y}(\alpha - y)}{\alpha^2 Z_b}, \tag{67}$$

the kernels of the linear integral operators $\hat{K}_b$ and $\hat{W}_b$ are defined by

$$K_b(q, q') = K(q, q') + \frac{\alpha}{\pi} e_-(q) e_-(q')(e^{iqy} + e^{iq'y}), \tag{68}$$

and

$$W_b(q, q') = -\frac{e_-(q) e_-(q') f(q) f(q')}{\pi \alpha^2 Z_b}, \tag{69}$$

respectively. The function $f$ is defined as

$$f(q) = \frac{2ie^{iqy} + \alpha e^{ik_- y}(q - k_+)}{q - k_-}, \tag{70}$$

$k_+$ and $k_-$ are defined by Eq. (31), and

$$Z_b = \frac{\delta_+ - \delta_- + 2\pi}{\alpha \pi}, \qquad \delta_\pm = \delta(\pm 1). \tag{71}$$

The other functions entering Eqs. (66)–(71) are defined in section 3.1. The typical behavior of the momentum distribution function is shown in Fig. 2(c).

# 4 Limit of strong interaction, $|\gamma| \to \infty$

Correlation functions of the model (1) in the $\gamma \to \infty$ limit has been represented as Fredholm determinants in the works [30, 31]. Using the Fredholm determinant representation we demonstrate that the one-body density matrix $\varrho(y)$ in the $\gamma \to \infty$ limit can be written as a correlation function of the one-dimensional impenetrable anyons. Such a correspondence remains valid for the gas state in the $\gamma \to -\infty$ limit.

## 4.1 Impurity-gas repulsion

We begin with discussing the $\gamma \to \infty$ limit of the impurity-gas repulsion. The kernels (61) and (62) simplify significantly when compared to arbitrary $\gamma$. Using that

$$\delta(q) = \frac{\pi}{2} - \arctan \Lambda + \frac{q\alpha}{1 + \Lambda^2} + \cdots, \qquad \alpha \to 0, \tag{72}$$

we have in the leading order in $\alpha$

$$Z = \frac{2\alpha}{1 + \Lambda^2}, \qquad \sin \delta(q) = \frac{1}{\sqrt{1 + \Lambda^2}}, \qquad e^{2i\delta(q)} = \frac{i\Lambda - 1}{i\Lambda + 1}, \qquad \alpha \to 0. \tag{73}$$

This gives us

$$K(q, q') = \frac{\lambda}{\pi} \frac{\sin[(q - q')y/2]}{q - q'}, \tag{74}$$

with

$$\lambda = \frac{2i}{\Lambda - i} \tag{75}$$

for the kernel (61), and

$$W(q, q') = \frac{e^{-iy(q+q')/2}}{2}, \tag{76}$$

for the kernel (62). The $\gamma \to \infty$ limit of Eq. (44) reads

$$Q = \frac{2 \arctan(\Lambda)}{\pi}. \tag{77}$$

Substituting this formula into Eq. (75) we get

$$\lambda = -1 - e^{-i\pi Q}. \tag{78}$$

Let us now show how $\varrho(y)$ emerges in the model of one-dimensional impenetrable anyons [32]. Recall that the anyon field operators satisfy the commutation relations

$$\psi_A(x_1)\psi_A^\dagger(x_2) = e^{-i\pi\kappa \operatorname{sgn}(x_1-x_2)}\psi_A^\dagger(x_2)\psi_A(x_1) + \delta(x_1 - x_2), \tag{79}$$

and

$$\psi_A^\dagger(x_1)\psi_A^\dagger(x_2) = e^{i\pi\kappa \operatorname{sgn}(x_1-x_2)}\psi_A^\dagger(x_2)\psi_A^\dagger(x_1). \tag{80}$$

Here, $\operatorname{sgn}(x) = |x|/x$, $\operatorname{sgn}(0) = 0$, and $\kappa$ is the statistics parameter. The correlation function $\langle\psi_A^\dagger(y)\psi_A(0)\rangle$ has the Fredholm determinant representation, given by Eq. (4) from Ref. [32]. The transformation explained in Ref. [5] (see the discussion of the equivalence between Eqs. (3.12) and (3.13) in Ch. XIII therein) leads us to the equality

$$\langle\psi_A^\dagger(y)\psi_A(0)\rangle = \varrho(y), \tag{81}$$

where $\lambda$ entering the kernel (74) is related to the statistical parameter $\kappa$ as follows:

$$\lambda = -1 - e^{i\pi\kappa}. \tag{82}$$

Comparing Eqs. (78) and (82) we get

$$\kappa = -Q \tag{83}$$

for $\kappa$ and $Q$ in the interval between minus one and one. The left hand side of Eq. (81) has also been extensively evaluated numerically [33,34]. However, no connection between the mobile impurity and anyon correlation functions, as suggested by Eqs. (81) and (83), has been given in the literature. Furthermore, the Jordan-Wigner transformation

$$\psi_A(x) = e^{-i\pi(1+\kappa)N(x)}\psi_F(x), \qquad N(x) = \int_{-\infty}^{x} dx'\, \psi_F^\dagger(x')\psi_F(x') \tag{84}$$

connects the anyon field operators and the fermion operators. Therefore, the right hand side of Eq. (81) is a correlation function of a free spinless Fermi gas:

$$\varrho(y) = \langle\mathrm{FS}|\psi_F^\dagger(y)e^{i\pi(\kappa+1)N(y)}e^{-i\pi(\kappa+1)N(0)}\psi_F(0)|\mathrm{FS}\rangle, \tag{85}$$

where $|\mathrm{FS}\rangle$ stands for the Fermi sea. Since Eq. (2.19) from Ch. XIII in Ref. [5] gives

$$\det(\hat{I} + \hat{K}) = \langle\mathrm{FS}|e^{i\pi(\kappa+1)N(y)}e^{-i\pi(\kappa+1)N(0)}|\mathrm{FS}\rangle, \tag{86}$$

it is $\psi_F^\dagger$ and $\psi_F$ that lead to the emergence of the rank-one operator $\hat{W}$ in Eq. (60). Note that the evaluation of the right hand side in Eqs. (85) and (86) can be done by using the Wick's theorem (for Eq. (86) see, e.g., Ref. [35]), without any use of the coordinate representation of the wave functions of the model. Interestingly, in a recent work [36] a two-dimensional impurity model has been linked to anyons, albeit in a different manner: There the statistical parameter is related to the impurity-phonon coupling.

### 4.2 Impurity-gas attraction: gas state

We now turn to the case of the gas state for the impurity-gas attraction, introduced in section 2.2. The $\gamma \to -\infty$ limit of the Fredholm determinant representation introduced in section (3.2), leads to the same formulas as the $\gamma \to \infty$ limit, discussed in section (4.1).

### 4.3 Impurity-gas attraction: bound state

Finally, we consider the bound state for the impurity-gas attraction. We take the $\gamma \to -\infty$ limit in the formulas of section (3.3) and get in the leading order

$$Z_b = \frac{2}{\alpha}, \qquad K_b(q,q') = K(q,q'), \qquad W_b(q,q') = 0, \qquad \gamma \to -\infty. \tag{87}$$

Furthermore, it follows from Eq. (51)

$$\Lambda = \frac{1}{2}\alpha Q, \qquad -1 \le Q \le 1, \qquad \gamma \to -\infty. \tag{88}$$

Therefore, we write the following asymptotic expression:

$$\varrho(y) = e^{y/\alpha}\left(1 - \frac{y}{\alpha}\right), \qquad \gamma \to -\infty. \tag{89}$$

Substituting this into Eq. (59) we get

$$n(k,Q) = \frac{2}{\pi}\frac{\alpha}{(1 + \alpha^2 k^2)^2}, \qquad \gamma \to -\infty. \tag{90}$$

The $\gamma \to \infty$ expansion (90) is not a uniform estimate of the exact result for $n(k,Q)$, since it misses the divergence at $k = Q$, discussed in detail in section 7. Still, it conveys an important message: the impurity momentum distribution becomes completely flat, and infinitely broad, in the $\gamma \to -\infty$ limit.

## 5 Total momentum $Q = 1 + 2 \times$ integer

The case

$$Q = 1 + 2 \times \text{integer} \tag{91}$$

is particular (recall that $k_F = 1$ everywhere but in the captions to the figures). One finds that $n(k,1)$ for the repulsive ground state, section 3.1, and the attractive gas state, section 3.2, coincide with the momentum distribution of a free Fermi gas. It follows from Eq. (44) that $\Lambda = \infty$ at $Q = 1$. We have in the leading order in $\Lambda$

$$\delta(k) = \frac{1}{\Lambda}, \qquad Z = \frac{2}{\pi\Lambda^2}, \qquad \Lambda \to \infty, \tag{92}$$

therefore

$$K(q,q') = 0, \qquad W(q,q') = \frac{1}{2}e^{-i(q+q')y/2} \tag{93}$$

and Eq. (60) takes the form

$$\varrho(y) = \frac{\sin(y)}{y}, \qquad \Lambda \to \infty. \tag{94}$$

Plugging this function into Eq. (59) we indeed get the momentum distribution of a free Fermi gas.

This result can also be obtained without using the Fredholm determinants. For the Gaudin-Yang model, Eq. (3), all three of the Hamiltonian, the spin-ladder operator

$$S_- = \int_0^L dx\, \psi_\downarrow^\dagger(x)\psi_\uparrow(x) = \sum_p \psi_{p\downarrow}^\dagger \psi_{p\uparrow}, \qquad (95)$$

and the total momentum $P$, commute with each other. Therefore, any state $|\Psi_{\text{FF}}\rangle$ of a free Fermi gas with $N+1$ particles can be turned into an eigenstate

$$|\Psi_i\rangle = \frac{1}{\sqrt{N+1}} S_- |\Psi_{\text{FF}}\rangle \qquad (96)$$

of the Hamiltonian (1), containing $N$ host particles and one impurity, and having the same energy and momentum as $|\Psi_{\text{FF}}\rangle$. Furthermore,

$$n_i(p, Q) \equiv \langle \Psi_i | \psi_{p\downarrow}^\dagger \psi_{p\downarrow} | \Psi_i \rangle = \frac{1}{N+1} \langle \Psi_{\text{FF}} | \psi_{p\uparrow}^\dagger \psi_{p\uparrow} | \Psi_{\text{FF}} \rangle. \qquad (97)$$

The state (96) is the minimum energy state $|\min_Q\rangle$ for $Q$ given by Eq. (91) and $|\Psi_{\text{FF}}\rangle$ being the minimum energy state of a free Fermi gas for the same $Q$. This can be shown very straightforwardly by examining the exact eigenfunctions and spectrum of the model (1), see, for example, section 5 of Supplementary Information in Ref. [37]. Equation (97) gives the momentum distribution of a free Fermi gas immediately.

The case of the bound state for the attractive interaction, sections 2.3 and 3.3, is different. The shape of $n(k, Q)$ is qualitatively the same at $Q$ given by Eq. (91) in comparison with any other value of $Q$. This is because the state (96) is not the minimum energy state of the Hamiltonian at any value of $Q$. We plot $n(k, 1)$ in Fig. 2(c).

# 6   $n(k, Q)$ in the $k \to \infty$ limit

The large $k$ limit of $n(k, Q)$, following Eq. (59), is determined by an expansion of $\varrho(y)$ in the vicinity of $y = 0$. It turns out that $\varrho$, $\partial_y \varrho$, and $\partial_y^2 \varrho$ are continuous at $y = 0$. Therefore

$$\langle P_{\text{imp}} \rangle \equiv \int dk\, k\, n(k, Q) = i\partial_y \varrho(y), \qquad y = 0 \qquad (98)$$

and

$$\langle P_{\text{imp}}^2 \rangle \equiv \int dk\, k^2 n(k, Q) = (i\partial_y)^2 \varrho(y), \qquad y = 0. \qquad (99)$$

The third derivative of $\varrho(y)$ has a discontinuity at $y = 0$. This implies for the leading term of the large $k$ expansion

$$n(k, Q) = \frac{1}{2\pi} \frac{1}{k^4} [\partial_y^3 \varrho(y = +0) - \partial_y^3 \varrho(y = -0)], \qquad k \to \pm\infty. \qquad (100)$$

Taking into account the involution (16) we arrive at

$$n(k, Q) = \frac{C}{k^4}, \qquad k \to \pm\infty, \qquad (101)$$

where

$$C = \frac{1}{\pi} \text{Re}\, \partial_y^3 \varrho(y = +0). \qquad (102)$$

Each of Eqs. (98), (99), and (101) has a lot of physics behind. We discuss them one-by-one.

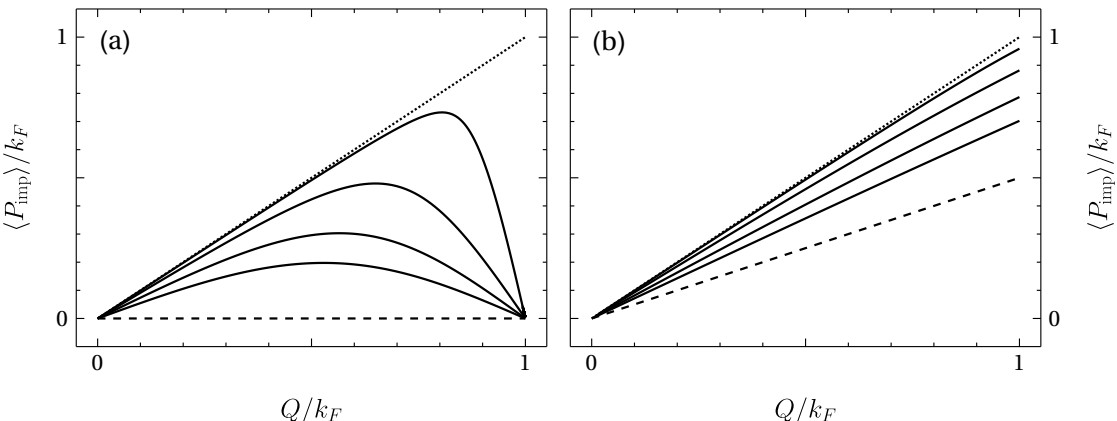

Figure 3: Average momentum $\langle P_{\mathrm{imp}} \rangle$ of the impurity is shown as a function of the total momentum $Q$. Panel (a) is for $\gamma > 0$ ground state, Eq. (103), panel (b) is for $\gamma < 0$ bound state, Eq. (104). The solid lines are for $|\gamma| = 1$, 3, 6, and 10 (top to bottom). Remarkably, they are continuous in (a), while experiencing a discontinuity in (b) at $Q = k_F$. They are not straight in (b), but this is barely seen with the unaided eye. The dotted and dashed lines stand for $\gamma = 0$ and $|\gamma| = \infty$, respectively, and are straight.

## 6.1 Analysis of $\langle P_{\mathrm{imp}} \rangle$

For the repulsive ground state, and the attractive gas state we have

$$\langle P_{\mathrm{imp}} \rangle = \frac{\Lambda}{\alpha} + \frac{\varphi}{Z}, \qquad \gamma > 0 \textbf{ ground state, and } \gamma < 0 \textbf{ gas state}, \tag{103}$$

where $Z$ is defined in Eq. (64) and $\varphi$ by Eq. (39). Recall that $\Lambda$ and $Q$ are connected by Eq. (44). Since $\langle P_{\mathrm{imp}} \rangle$ in Eq. (103) is an odd function of $\gamma$, it is sufficient to examine the $\gamma > 0$ case. For the attractive bound state $Z$ is replaced with $Z_b$, Eq. (71). Hence,

$$\langle P_{\mathrm{imp}} \rangle = \frac{\Lambda}{\alpha} + \frac{\varphi}{Z_b}, \qquad \gamma < 0 \textbf{ bound state}, \tag{104}$$

where $\Lambda$ and $Q$ are connected by Eq. (51). Using the Hellmann-Feynman theorem as explained in Ref. [38] gives the average momentum of the impurity in terms of the group velocity,

$$\langle P_{\mathrm{imp}} \rangle = \frac{\partial E_{\min}(Q)}{\partial Q}. \tag{105}$$

This leads us to Eqs. (103) and (104) immediately, consistent with the predictions from the Fredholm determinant representation.

The derivation of Eqs. (103) and (104) from the Fredholm determinant representation of Eq. (98) is performed in Appendix B. Though Eqs. (60) and (66) look rather different, Eq. (104) is connected to Eq. (103) by merely a replacement of $Z$ with $Z_b$. Notably, such a replacement also works for the other observables considered in section 6: $\langle P_{\mathrm{imp}}^2 \rangle$, Eq. (99), and $C$, Eq. (102). We show $\langle P_{\mathrm{imp}} \rangle$ for several values of $\gamma$ in Fig. 3. One can see in this figure that Eq. (103) produces a continuous function of $Q$, while Eq. (104) exhibits a discontinuity at $Q = 1$. Should such a difference persist for any one-dimensional gas interacting with a mobile impurity, is an open question.

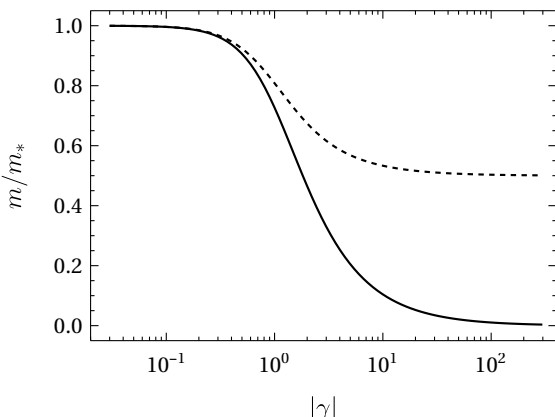

Figure 4: The ratio of the impurity's bare and effective masses, $m/m_*$. The solid line is for the $\gamma > 0$ ground state, Eq. (108), and the dashed line is for the $\gamma < 0$ bound state, Eq. (109). The former line tends to zero, and the latter tends to $1/2$, in the $|\gamma| \to \infty$ limit.

Though the curves in Fig. 3(b) are not straight lines, the difference cannot be seen by a naked eye. It follows from Eq. (105) that the slope of $\langle P_{\text{imp}} \rangle$ at $Q = 0$,

$$\langle P_{\text{imp}} \rangle = \frac{Q}{m_*}, \qquad Q \to 0 \tag{106}$$

is set by the value of the effective mass $m_*$ defined by the expansion of $E(Q)$ at $Q = 0$:

$$E(Q) - E(0) = \frac{Q^2}{2m_*}, \qquad Q \to 0. \tag{107}$$

The explicit form of $E(Q)$ is discussed in section 2. The analytic formula for $m_*$ corresponding to Eq. (103) is

$$m_* = \frac{2}{\pi} \frac{(\arctan \alpha)^2}{\arctan \alpha - \alpha(1 + \alpha^2)^{-1}}, \qquad \gamma > 0 \text{ ground state, and } \gamma < 0 \text{ gas state} \tag{108}$$

(note that $m_*$ in this equation is an odd function of $\gamma$), and the formula for $m_*$ corresponding to Eq. (104) is

$$m_* = \frac{2}{\pi} \frac{(\pi + \arctan \alpha)^2}{\pi + \arctan \alpha - \alpha(1 + \alpha^2)^{-1}}, \qquad \gamma < 0 \text{ bound state}. \tag{109}$$

The analytic expressions (108) and (109) for the effective mass were obtained for the first time in the works [16] and [17], respectively. The $\gamma \to \infty$ limit of Eq. (108) is $m/m_* = 0$: the impurity becomes infinitely heavy. This is contrasted with the $\gamma \to -\infty$ limit of Eq. (109), which is $m/m_* = 1/2$: the mass of the impurity bound to the gas particles remains finite. A quantitative comparison between $m_*$ for $\gamma > 0$ from Eq. (108), and $m_*$ for $\gamma < 0$ from Eq. (109) is made in Fig. 4 .

## 6.2 Analysis of the coefficient $C$ in the large $k$ expansion $n(k, Q) = C/k^4$

In this section we give the explicit analytic formula for the coefficient $C$ in Eq. (102). For the repulsive ground state, and the attractive gas state we have

$$C = \frac{1}{\pi}\left( \frac{2}{\pi\alpha^2} - \frac{Z}{\alpha^2} - \frac{\varphi^2}{Z} \right), \qquad \gamma > 0 \text{ ground state, and } \gamma < 0 \text{ gas state}, \tag{110}$$

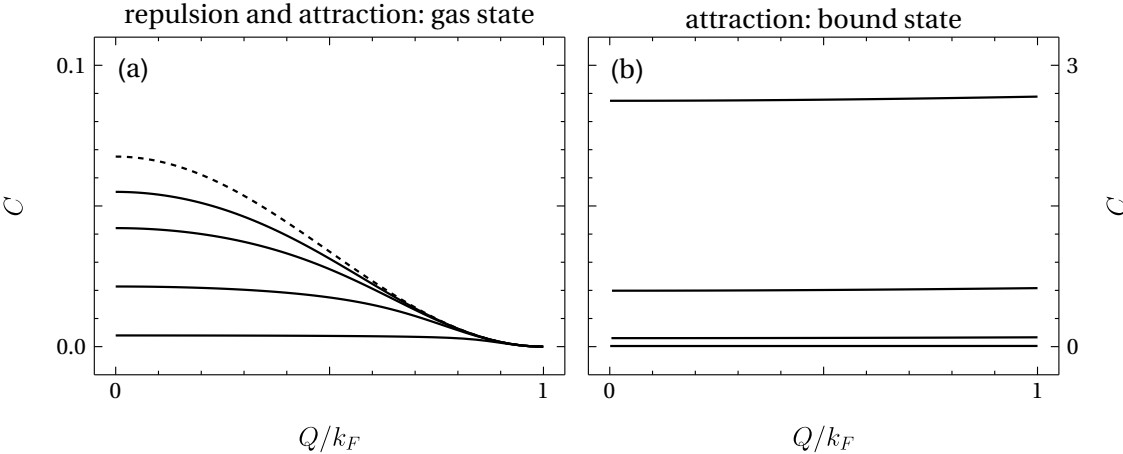

Figure 5: The contact $C$ as a function of the total momentum $Q$. Panel (a) is for $\gamma > 0$ ground state (identical to $\gamma < 0$ gas state), Eq. (110). Panel (b) is for $\gamma < 0$ bound state, Eq. (111). The solid lines are for $|\gamma| = 1, 3, 6$, and $10$ (bottom to top). The lines in (b) are not straight, but this is barely seen with the unaided eye. The dashed line in (a) is $C = 2[\cos(\pi Q/2k_F)]^2/3\pi^2$, given for $|\gamma| = \infty$ by Eq. (112).

where $Z$ is defined in Eq. (64) and $\varphi$ by Eq. (39). Recall that $\Lambda$ and $Q$ are connected by Eq. (44), and note that $C$ in Eq. (110) is an even function of $\gamma$. For the attractive bound state $Z$ is replaced with $Z_b$, Eq. (71). Hence,

$$C = \frac{1}{\pi}\left(\frac{2}{\pi\alpha^2} - \frac{Z_b}{\alpha^2} - \frac{\varphi^2}{Z_b}\right), \qquad \gamma < 0 \text{ \textbf{bound state}}. \tag{111}$$

The $\gamma \to \infty$ limit of Eq. (110) reads

$$C = \frac{2}{3\pi^2}\left[\cos\left(\frac{\pi Q}{2}\right)\right]^2, \qquad |\gamma| \to \infty. \tag{112}$$

The $\gamma \to -\infty$ limit of Eq. (111) is divergent, in consistency with the analysis of section 4.3. We show $C$ for several values of $\gamma$ in Fig. 5.

The case $Q = 0$ can be compared with the existing literature. Equations (110) and (111) become

$$C = \frac{2(\alpha - \arctan\alpha)}{\pi^2\alpha^3}, \qquad Q = 0, \qquad \gamma > 0 \text{ \textbf{ground state, and} } \gamma < 0 \text{ \textbf{gas state}}, \tag{113}$$

and

$$C = \frac{2(-\pi + \alpha - \arctan\alpha)}{\pi^2\alpha^3}, \qquad Q = 0, \qquad \gamma < 0 \text{ \textbf{bound state}}, \tag{114}$$

respectively. One can check that

$$C = \frac{\gamma^2}{2\pi^2}\frac{\partial E_{\min}}{\partial\gamma}, \qquad Q = 0, \tag{115}$$

where $E_{\min}$ is given by Eqs. (46) and (54), respectively. This result is consistent with the general principles determining the coefficient $C$ (sometimes referred to as the contact), developed in the works [39–42]. Notably, the contact in the Lieb-Liniger gas [43] has the value $2/(3\pi^2)$ in the Tonks-Girardeau limit. This coincides with what gives Eq. (112) at $Q = 0$.

To what extent $C$ could be extracted numerically from the large momentum behavior of $n(k, Q)$ is illustrated in Fig. (6). We evaluated $n(k, Q)$ from the Fredholm determinant representation presented in section 3.

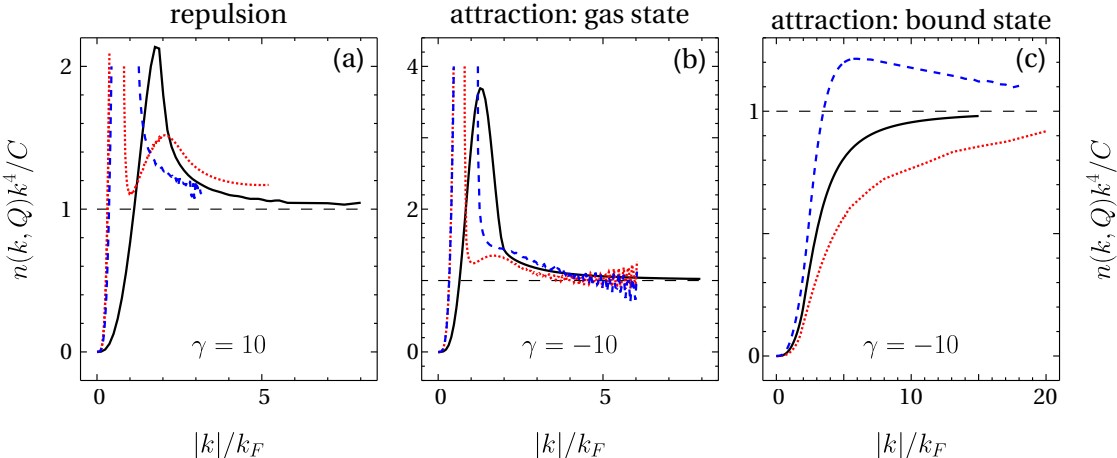

Figure 6: Shown is the convergence of $n(k,Q)$ to $Ck^{-4}$ in the large $k$ limit. The plots are from the numeric evaluation of the Fredholm determinant representation for $n(k,Q)$ given in section 3, divided by the value of $C$ found analytically in section 6.2. The solid black lines are for $Q = 0$. The dotted red, and dashed blue lines are for $Q = 0.8k_F$: the former is for $k > 0$, and the latter is for $k < 0$. Note that the Fredholm determinants are numerics-friendly, but $n(k,Q)$ decays very fast with increasing $|k|$, and this makes the numerical evaluation of $C$ a challenge.

## 6.3 Analysis of $\langle P_{\text{imp}}^2 \rangle$

The average of $P_{\text{imp}}^2$, Eq. (99), is expressed through $\langle P_{\text{imp}} \rangle$ and $C$:

$$\sigma = \sqrt{\pi C(Z^{-1} - \alpha)}, \tag{116}$$

where, by definition,

$$\sigma = \sqrt{\langle P_{\text{imp}}^2 \rangle - \langle P_{\text{imp}} \rangle^2} \tag{117}$$

is a root-mean-square deviation. Equation (116) is valid for the repulsive ground state and attractive gas state. The result for the attractive bound state is obtained by replacing $Z$ with $Z_b$. Exemplary plots of $\sigma$ are shown in Fig. 7.

## 7 $n(k,Q)$ in the $k \to Q$ limit

In this section we present the $y \to \infty$ expansion of $\varrho(y)$. We use it to prove the existence of the power-law singularity

$$n(k,Q) \sim \frac{1}{(k-Q)^\nu}, \qquad k \to Q, \tag{118}$$

seen in Fig. 2, as well as to calculate the exponent $\nu$, and the numerical prefactor. So far, $\nu$ has only been found at $Q = 0$ and $\gamma \to +0$ in Ref. [12]; this result follows from our formulas as a particular case.

### 7.1 Large $y$ expansion of $\varrho(y)$ in case of impurity-gas repulsion

The density matrix and the momentum distribution are related by Eq. (59). Both are $2k_F$-periodic in $Q$ (recall that $k_F = 1$ everywhere but in the captions to the figures). This property together with Eq. (17) makes it sufficient to examine $\varrho$ for $0 \leq Q \leq 1$ only. The large $y$

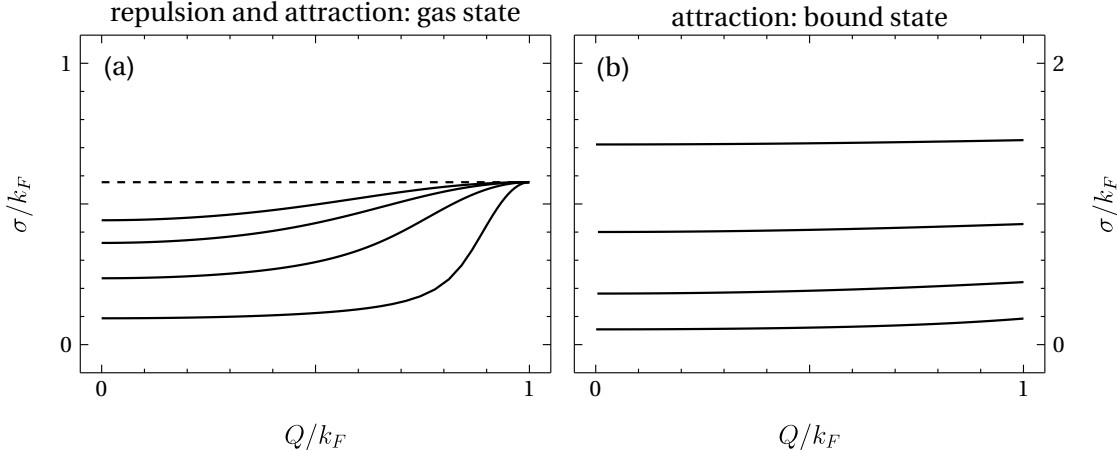

Figure 7: The root-mean-square deviation $\sigma$ as a function of the total momentum $Q$. Panel (a) is for $\gamma > 0$ ground state (identical to $\gamma < 0$ gas state), Eq. (116). Panel (b) is for $\gamma < 0$ bound state. The solid lines are for $|\gamma| = 1, 3, 6,$ and $10$ (bottom to top). The lines in (b) are not straight, but this is barely seen with the unaided eye. The horizontal dashed line at $\sigma/k_F = 1/\sqrt{3}$ in (a) is for $|\gamma| = \infty$.

expansion of the determinant representation (60) can be obtained by a finite-size analysis of the form-factors followed by a resummation of the soft modes, along the lines of the works [15, 44–48]. We leave the details for a separate publication. The result is

$$\varrho(y) = \frac{\mathcal{A}e^{-iQy}}{(2iy)^{F_-^2}(-2iy)^{(1-F_+)^2}} + \frac{\tilde{\mathcal{A}}e^{-i(Q-2)y}}{(2iy)^{\tilde{F}_-^2}(-2iy)^{(1-\tilde{F}_+)^2}} + \cdots, \qquad y \to \infty. \tag{119}$$

The numerical prefactor

$$\mathcal{A} = (2\pi)^{F_- - F_+ + 1} e^{-\Delta} Z^{-1} G^2(F_+) G^2(1 - F_-) \tag{120}$$

depends on $\gamma$ and $Q$ through the phase shift (65):

$$F(k) = \frac{\delta(k)}{\pi}, \qquad F_\pm = F(\pm 1). \tag{121}$$

Here,

$$\Delta = \frac{1}{2} \int_{-1}^{1} dq \int_{-1}^{1} dq' \left[ \frac{F(q) - F(q')}{q - q'} \right]^2 + \int_{-1}^{1} dq \frac{F_-^2 - F^2(q)}{-1 - q} - \int_{-1}^{1} dq \frac{(1 - F_+)^2 - [1 - F(q)]^2}{1 - q}, \tag{122}$$

the coefficient $Z$ is given by Eq. (64):

$$Z = \frac{F_+ - F_-}{\alpha}, \tag{123}$$

and $G$ stands for the Barnes $G$-function, defined by the functional equation

$$G(z + 1) = \Gamma(z) G(z), \tag{124}$$

with the normalization $G(1) = 1$, where $\Gamma(z)$ is the Euler Gamma function. The function $\tilde{F}$ entering the second term on the right hand side of Eq. (119) is

$$\tilde{F}(k) = F(k) + 1, \tag{125}$$

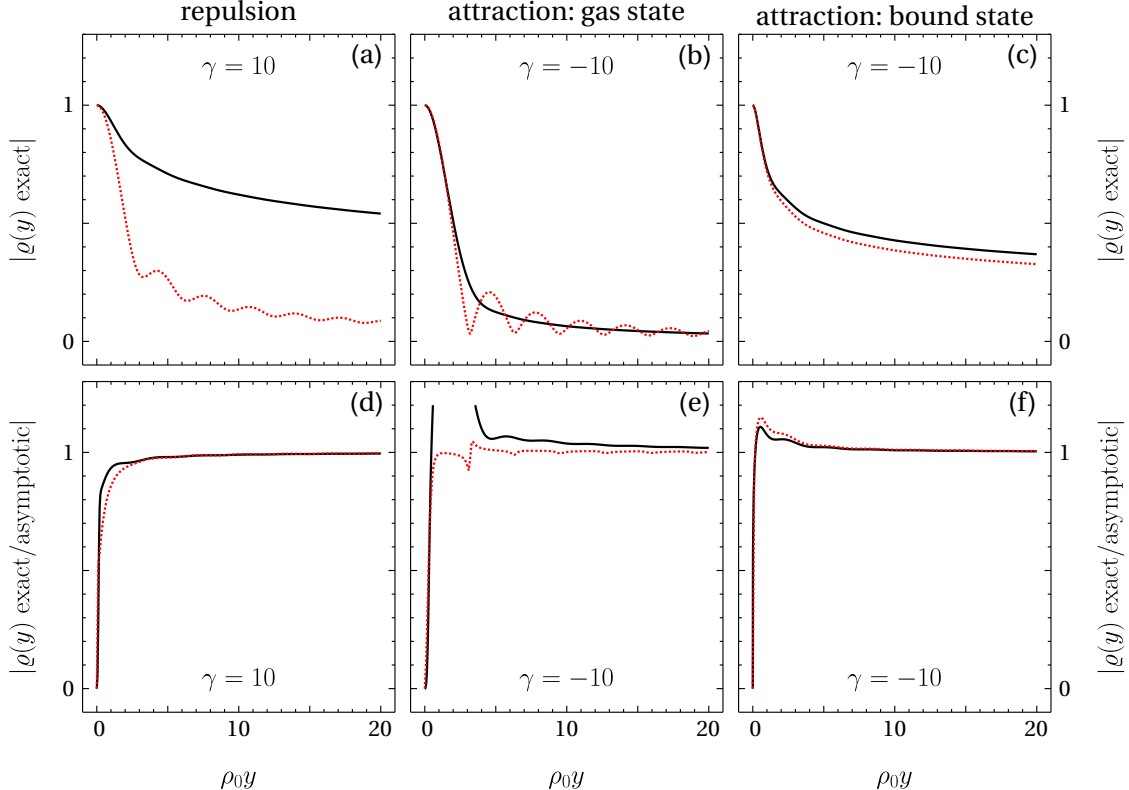

Figure 8: One-particle density matrix $\varrho(y)$ at $Q = 0$ (black solid) and $Q = 0.8k_F$ (red dotted) lines. Top panels: absolute value of $\varrho(y)$ from the exact formula. Bottom panels: the absolute value of the ratio of $\varrho(y)$ from the exact and large-$y$-asymptotic formulas.

and $\tilde{\mathcal{A}}$ follows from $\mathcal{A}$ by replacing $F$ with $\tilde{F}$ in Eqs. (120), (122) and (123). The second term on the right hand side of Eq. (119) is, generally, subleading – it decays faster than the first one:

$$\tilde{F}_-^2 + (1 - \tilde{F}_+)^2 \geq F_-^2 + (1 - F_+)^2. \tag{126}$$

However, the inequality turns into an equality at $Q = 1$, that is, the subleading term becomes of the same order as the leading one, and their sum in Eq. (119) reproduces the exact formula

$$\varrho(y) = \frac{\sin y}{y}, \qquad Q = 1. \tag{127}$$

We show $\varrho(y)$ evaluated from the exact expression (60), and the convergence of the asymptotic formula (119) to this exact expression in the panels (a) and (d) of Fig. (8), respectively. We would like to emphasize that the decay rates of the leading and the first subleading terms in Eq. (119) are close to each other when $Q$ is close to one.

## 7.2 Large $y$ expansion of $\varrho(y)$ in case of impurity-gas attraction: gas state

All formulas from the section 7.1 are valid for the gas state after letting $\gamma$ be negative. We show $\varrho(y)$ evaluated from the exact expression (60), and the convergence of the asymptotic formula (119) to this exact expression in the panels (b) and (e) of Fig. (8), respectively.

### 7.3 Large $y$ expansion of $\varrho(y)$ in case of impurity-gas attraction: bound state

In case of the attractive bound state, the explicit expression for $\varrho(y)$ is given by Eq. (66), and the leading term in the $y \to \infty$ expansion reads

$$\varrho(y) = \frac{\mathcal{A}_b e^{-iQy}}{(2iy)^{(1-F_-)^2}(-2iy)^{F_+^2}}, \qquad y \to \infty, \tag{128}$$

where

$$\mathcal{A}_b = \frac{8(2\pi)^{F_- - F_+}}{\pi |Z_b|} \frac{G^2(1+F_+)G^2(2-F_-)}{[1+(\alpha+\Lambda)^2]^2} e^{-\Delta_b}, \tag{129}$$

with

$$\Delta_b = \frac{1}{2}\int_{-1}^{1} dq \int_{-1}^{1} dq' \left[\frac{F(q)-F(q')}{q-q'}\right]^2 + \int_{-1}^{1} dq \frac{(1-F_-)^2 - [1-F(q)]^2}{-1-q}$$

$$- \int_{-1}^{1} dq \frac{F_+^2 - F(q)^2}{1-q} - 4\alpha \int_{-1}^{1} dq \frac{F(q)(\Lambda - \alpha q)}{1+(\Lambda-\alpha q)^2}, \tag{130}$$

and $Z_b$ given by Eq. (71). The prefactors $\mathcal{A}$ and $\tilde{\mathcal{A}}$, Eq. (120), depend on $\gamma$ and $Q$ through the phase shift only. By contrast, the prefactor $\mathcal{A}_b$, Eq. (129), depends on $\gamma$ and $Q$ explicitly.

We show $\varrho(y)$ evaluated from the exact expression (66), and the convergence of the asymptotic formula (128) to this exact expression in the panels (c) and (f) of Fig. (8), respectively.

### 7.4 The exponent $\nu$ and the prefactor in Eq. (118) for $n(k,Q)$

The singular part of the momentum distribution, Eq. (118), is fully characterized by the asymptotic expressions for $\varrho(y)$. Equation (119) leads to the exponent

$$\nu = 1 - F_-^2 - (1-F_+)^2, \qquad \gamma > 0 \text{ ground state, and } \gamma < 0 \text{ gas state}, \tag{131}$$

and Eq. (128) leads to

$$\nu = 1 - (1-F_-)^2 - F_+^2, \qquad \gamma < 0 \text{ bound state}. \tag{132}$$

Both Eqs. (131) and (132) tend to the same value in the $|\gamma| \to \infty$ limit,

$$\nu = \frac{1-Q^2}{2}, \qquad |\gamma| \to \infty, \tag{133}$$

which coincides with the result from Ref. [23]. This limiting value is indicated with the thin dotted line in Fig. 9. One can also see that $\nu = 0$ when $Q$ reaches the Fermi momentum for the $\gamma > 0$ ground state, and $\gamma < 0$ gas state. Recall that $n(k,Q)$ turns into the Fermi function at $Q = 1$, as illustrated in the panels (a) and (b) of Fig. 2 and discussed in section 5. The case $\gamma < 0$ bound state is different, there $\nu$ is a non-trivial function of $\gamma$ at $Q = 1$.

Letting $Q = 0$ and $\gamma \to +0$ in Eq. (131) we get

$$\nu = 1 - \frac{\gamma^2}{2\pi^4} + \cdots, \qquad Q = 0, \qquad \gamma \to +0. \tag{134}$$

This gives the same dependence on $\gamma$ as in Ref. [12].

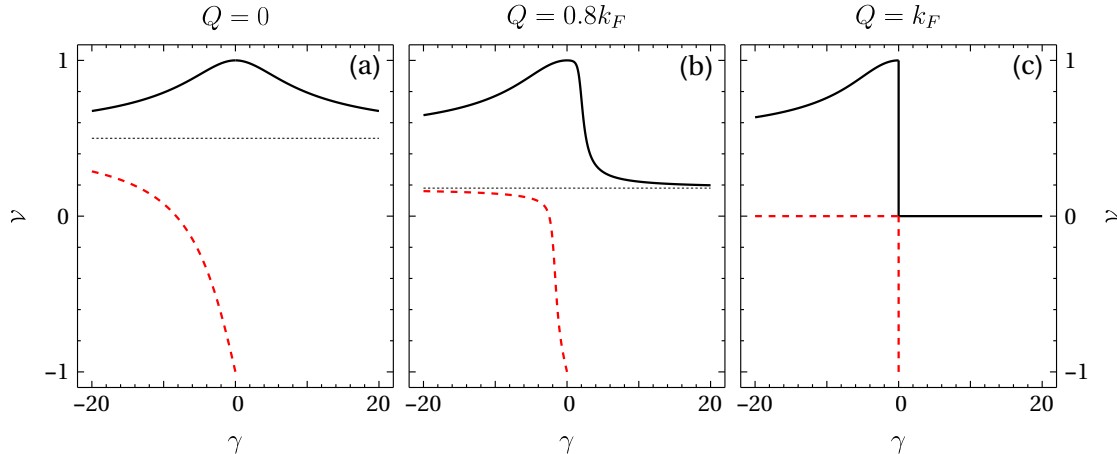

Figure 9: Exponent $\nu$ for the singularity $n(k,Q) \sim (k-Q)^{-\nu}$ in the $k \to Q$ limit is shown as a function of $\gamma$. Solid line is for $\gamma > 0$ ground state and $\gamma < 0$ bound state, dashed red line is for $\gamma < 0$ gas state. Thin dotted line indicates where $\nu$ tends in the $|\gamma| \to \infty$ limit.

# 8  Determinant representation for finite $N$

In this section we present the impurity momentum distribution function $n(k,Q)$ for a finite particle number $N$ through determinants of finite-dimensional matrices. This result is crucial for deriving the Fredholm determinant representation of section 3. Recall that we stick to the notations of the paper [25], whenever possible.

Our starting point is Eq. (19). We write the form-factor as given by Eq. (5.23) from Ref. [25]:

$$|\langle N|\psi_{k\downarrow}|\min_Q\rangle|^2 = \left(\frac{2}{L}\right)^N |\det D|^2 \left|\sum_{j=1}^{N+1} \frac{\partial k_j}{\partial \Lambda}\right|^{-1} \left|\prod_{j=1}^{N+1} \frac{\partial k_j}{\partial \Lambda}\right|. \tag{135}$$

Here, $\partial k_j/\partial \Lambda$ is defined by Eq. (32), and

$$\det D = \begin{vmatrix} \dfrac{1}{k_1-p_1} & \cdots & \dfrac{1}{k_{N+1}-p_1} \\ \vdots & \ddots & \vdots \\ \dfrac{1}{k_1-p_N} & \cdots & \dfrac{1}{k_{N+1}-p_N} \\ 1 & \cdots & 1 \end{vmatrix} \tag{136}$$

for the determinant of the $(N+1)\times(N+1)$ matrix. The momentum $Q$ of the state $|\min_Q\rangle$ is the sum of the quasi-momenta $k_1,\ldots,k_{N+1}$, Eq. (24). How these quasi-momenta are specified is discussed in sections 2.1 through 2.3. The momentum of the state $|N\rangle$ is the sum of $p_1,\ldots,p_N$. Combining Eqs. (21) and (24) implies the constraint

$$k + \sum_{j=1}^{N} p_j = \sum_{j=1}^{N+1} k_j \tag{137}$$

for the sum over $p_1,\ldots,p_N$ in Eq. (19).

We transform Eq. (19) by replacing the constraint (137) with the Kronecker delta:

$$n(k,Q) = \frac{1}{N!} \sum_{p_1} \cdots \sum_{p_N} \delta_{k+\sum_{j=1}^{N} p_j, \sum_{j=1}^{N+1} k_j} |\langle N|\psi_{k\downarrow}|\min_Q\rangle|^2. \tag{138}$$

The summations over $p_1, \ldots, p_N$ on the right hand side of Eq. (138) run independently from each other. One can see from Eqs. (135) and (136) that $\langle N|\psi_{k\downarrow}|\min_Q\rangle = 0$ if $p_j = p_l$ at $j \neq l$. The factor $1/N!$ is to compensate counting the form-factor multiple times upon the permutations of $p_1, \ldots, p_N$. Equations (12) and (16), and the representation

$$\delta_{k+\sum_{j=1}^{N} p_j, \sum_{j=1}^{N+1} k_j} = \frac{1}{L} \int\limits_{-L/2}^{L/2} dy \, \exp\left[ iy \left( k + \sum_{j=1}^{N} p_j - \sum_{j=1}^{N+1} k_j \right) \right] \tag{139}$$

imply for Eq. (138)

$$n(k,Q) = \frac{1}{L} \int\limits_{-L/2}^{L/2} dy \, e^{iky} \varrho(y) = \frac{2}{L} \int\limits_{0}^{L/2} dy \, \text{Re}[e^{iky} \varrho(y)], \tag{140}$$

where

$$\varrho(y) = \frac{1}{N!} \sum_{p_1} \cdots \sum_{p_N} e^{iy\left(\sum_{j=1}^{N} p_j - \sum_{j=1}^{N+1} k_j\right)} |\langle N|\psi_{k\downarrow}|\min_Q\rangle|^2. \tag{141}$$

The terms on the right hand side of Eq. (141) are determined by Eq. (135), and $p_1, \ldots, p_N$ are quantized as given by Eq. (22).

We now take the sum over $p_1, \ldots, p_N$ in Eq. (141). Let us consider the function

$$S = \frac{1}{N!} \sum_{p_1} \cdots \sum_{p_N} (\det D)^2 \prod_{j=1}^{N} f(p_j), \tag{142}$$

where $\det D$ is defined by Eq. (136), $f$ is an arbitrary function, and $p_j$s are quantized as given by Eq. (22). After some elementary transformations (used, for example, to get the identities in appendix B.3 from Ref. [25]) we come at the following representation for Eq. (142):

$$S = \sum_{m=1}^{N+1} \det[\alpha(m)_{jl}]. \tag{143}$$

Here,

$$\alpha(m)_{jl} = \begin{cases} \displaystyle\sum_{p} \frac{f(p)}{(k_j - p)(k_l - p)} & 1 \leq j \neq m \leq N+1, \\[4mm] 1 & j = m, \end{cases} \tag{144}$$

and $p = 2\pi n/L$, $n = 0, \pm 1, \pm 2, \ldots$.

For $\gamma > 0$ repulsive ground state and $\gamma < 0$ attractive gas state the quasi-momenta $k_1, \ldots, k_{N+1}$ are real. This implies

$$|\det D|^2 = (\det D)^2. \tag{145}$$

Furthermore, one can show that

$$\frac{\partial k_j}{\partial \Lambda} > 0, \qquad -\infty < \Lambda < \infty, \qquad j = 1, \ldots N+1 \tag{146}$$

for any real-valued $k_j$ (see, for example, section 5.2 from Ref. [25]). We, therefore, can use the identity (143) for the function (141), and get

$$\varrho(y) = \partial_\xi \det(A + \xi B)|_{\xi=0}, \tag{147}$$

where

$$A_{jl} = \frac{2}{L} \sum_n \frac{e^{2\pi i y n/L}}{(k_j - 2\pi n/L)(k_l - 2\pi n/L)} e^{-iy(k_j+k_l)/2} \left|\frac{\partial k_j}{\partial \Lambda}\right|^{1/2} \left|\frac{\partial k_l}{\partial \Lambda}\right|^{1/2} \tag{148}$$

and

$$B_{jl} = \left(\sum_{m=1}^{N+1} \frac{\partial k_m}{\partial \Lambda}\right)^{-1} e^{-iy(k_j+k_l)/2} \left|\frac{\partial k_j}{\partial \Lambda}\right|^{1/2} \left|\frac{\partial k_l}{\partial \Lambda}\right|^{1/2}. \tag{149}$$

The matrix $B$ has rank one, and we can write Eq. (147) as

$$\varrho(y) = \det(A+B) - \det A. \tag{150}$$

We now turn to the $\gamma < 0$ bound state. Here, $k_1, \ldots, k_{N-1}$ are real, and $k_N = k_{N+1}^*$ are complex. This implies

$$|\det D|^2 = -(\det D)^2. \tag{151}$$

It follows from Eq. (24) that

$$\sum_{j=1}^{N+1} \frac{\partial k_j}{\partial \Lambda} = \frac{\partial Q}{\partial \Lambda}. \tag{152}$$

Since $Q$ and $\Lambda$ are connected by Eq. (51), we get

$$\sum_{j=1}^{N+1} \frac{\partial k_j}{\partial \Lambda} = -\left|\sum_{j=1}^{N+1} \frac{\partial k_j}{\partial \Lambda}\right| < 0. \tag{153}$$

Using the identity (143) for the function (141) we come at Eqs. (147)–(149).

Later, we will use the following representation for the entries of the matrix (148):

$$A_{jl} = -\frac{c(k_j) - c(k_l)}{k_j - k_l} e^{-iy(k_j+k_l)/2} \left|\frac{\partial k_j}{\partial \Lambda}\right|^{1/2} \left|\frac{\partial k_l}{\partial \Lambda}\right|^{1/2}, \tag{154}$$

where

$$c(k) = \frac{2}{L} \sum_n \frac{e^{2\pi i y n/L}}{k - 2\pi n/L}. \tag{155}$$

The uncertainty in Eq. (154) at $j = l$ can be resolved by L'Hôpital's rule, which amounts to making use of the expansion

$$c(k_l) = c(k_j) + (k_l - k_j) \left.\frac{\partial c(k)}{\partial k}\right|_{k=k_j}. \tag{156}$$

That is,

$$A_{jj} = -e^{-iyk_j} \left|\frac{\partial k_j}{\partial \Lambda}\right| \left.\frac{\partial c(k)}{\partial k}\right|_{k=k_j}, \tag{157}$$

where $c(k)$ is given by Eq. (155) and $\partial k_j/\partial \Lambda$ by Eq. (32).

Let us represent the function $c$ from Eq. (155) as

$$c(k) = \oint_\Gamma \frac{dz}{\pi} \frac{e^{izy}}{e^{iLz} - 1} \frac{1}{k - z}, \tag{158}$$

where $\Gamma$ is a union of counter-clockwise-oriented contours around the points $z = 2\pi n/L$. Assuming that $k$ is real, we deform $\Gamma$ into a contour encircling the point $z = k$, and two straight lines infinitesimally above and below the real axis:

$$c(k) = 2i \frac{e^{iky}}{e^{iLk} - 1} - \int_{-\infty+i0}^{\infty+i0} \frac{dz}{\pi} \frac{e^{izy}}{e^{iLz} - 1} \frac{1}{k - z} + \int_{-\infty-i0}^{\infty-i0} \frac{dz}{\pi} \frac{e^{iz(y-L)}}{1 - e^{-iLz}} \frac{1}{k - z}. \tag{159}$$

We assume $0 < y < L$; the result for $y = 0$ and $y = L$ follows from the continuity of $\varrho(y)$. The first integral is equal to zero, which is seen by using Cauchy's residue theorem (the integration contour is extended to the closed one by adding a half-circle in the upper half-plane). The second integral is equal to zero for the same reason (the integration contour is extended to the lower half-plane). Therefore, we get for Eq. (155):

$$c(k) = 2i\frac{e^{iky}}{e^{ikL} - 1}. \tag{160}$$

We now introduce the function

$$e(k) = \frac{e^{iky}}{v_-(k)}. \tag{161}$$

Substituting the Bethe equations (29) into Eq. (160) we find

$$c(k_j) = e(k_j), \qquad j = 1, \ldots, N + 1. \tag{162}$$

Furthermore,

$$\frac{\partial c(k)}{\partial k} = ic(k)\left(y - \frac{L}{1 - e^{-ikL}}\right), \tag{163}$$

and

$$\frac{\partial e(k)}{\partial k} = ie(k)[y - i\alpha v_-(k)]. \tag{164}$$

Using Eqs. (161)–(163) we get for Eq. (157)

$$A_{jj} = -i\left|\frac{\partial k_j}{\partial \Lambda}\right|\frac{1}{v_-(k_j)}\left[y - \frac{L}{2i}\frac{1}{v_+(k_j)}\right]. \tag{165}$$

This expression can be represented as follows

$$A_{jj} = 1 - e^{-ik_j y}\left|\frac{\partial k_j}{\partial \Lambda}\right|\left.\frac{\partial e(k)}{\partial k}\right|_{k=k_j}. \tag{166}$$

Thus, we can write the matrix (154) as

$$A_{jl} = \delta_{jl} - \frac{e(k_j) - e(k_l)}{k_j - k_l}e^{-iy(k_j+k_l)/2}\left|\frac{\partial k_j}{\partial \Lambda}\right|^{1/2}\left|\frac{\partial k_l}{\partial \Lambda}\right|^{1/2}. \tag{167}$$

Equation (166) can be obtained from Eq. (167) by making use of the L'Hôpital's rule.

Let us represent Eq. (167) as

$$A_{jl} = \delta_{jl} + \frac{2\pi}{L}K(k_j, k_l), \qquad j, l = 1, \ldots, N + 1, \tag{168}$$

where

$$K(k_j, k_l) = \frac{e_+(k_j)e_-(k_l) - e_-(k_j)e_+(k_l)}{k_j - k_l}, \qquad j, l = 1, \ldots, N + 1. \tag{169}$$

Here,

$$e_+(k_j) = -\frac{1}{\pi}\frac{e^{ik_j y/2}}{v_-(k_j)}\left|\frac{L}{2}\frac{\partial k_j}{\partial \Lambda}\right|^{1/2}, \qquad e_-(k_j) = e^{-ik_j y/2}\left|\frac{L}{2}\frac{\partial k_j}{\partial \Lambda}\right|^{1/2}, \tag{170}$$

where $\partial k_j/\partial \Lambda$ is defined by the exact formula (32). The uncertainty in Eq. (169) at $j = l$ can be resolved by L'Hôpital's rule. The matrix (149) can be written as

$$B_{jl} = \frac{2\pi}{L}W(k_j, k_l), \qquad j, l = 1, \ldots, N + 1, \tag{171}$$

where

$$W(k_j, k_l) = \frac{1}{\pi}\left(\sum_{m=1}^{N+1}\frac{\partial k_m}{\partial \Lambda}\right)^{-1} e_-(k_j)e_-(k_l), \qquad j, l = 1, \ldots, N+1. \tag{172}$$

Using Eqs. (168)–(172) we get for Eq. (150)

$$\varrho(y) = \det\left[\delta_{jl} + \frac{2\pi}{L}K(k_j, k_l) + \frac{2\pi}{L}W(k_j, k_l)\right] - \det\left[\delta_{jl} + \frac{2\pi}{L}K(k_j, k_l)\right]. \tag{173}$$

Recall that we are working at a finite constant density, Eq. (9). The expression (173) is valid in the interval $0 \leq y \leq L$.

That the exact function $\varrho(y)$ is $L$-periodic and satisfies the involution (16) implies the exact identity

$$\varrho(L - y) = \varrho^*(y). \tag{174}$$

We have verified numerically that Eq. (173) with the kernels (169)–(172) satisfies Eq. (174) for any $N$, and $y$ in the interval $0 \leq y \leq L$. We have also verified it by performing symbolic computations using MATHEMATICA package for $N = 2$.

Let us now discuss the case of the complex quasi-momenta: $\mathrm{Im}(k_N) < 0$ and $\mathrm{Im}(k_{N+1}) > 0$, Eq. (49). The representation (158) leads to

$$c(k) = -\int_{-\infty+i0}^{\infty+i0}\frac{dz}{\pi}\frac{e^{izy}}{e^{iLz}-1}\frac{1}{k-z} + \int_{-\infty-i0}^{\infty-i0}\frac{dz}{\pi}\frac{e^{izy}}{e^{iLz}-1}\frac{1}{k-z}. \tag{175}$$

The first (second) integral gives non-zero contribution for $\mathrm{Im}(k) > 0$ ($\mathrm{Im}(k) < 0$). In both cases one arrives at Eq. (160). Further analysis is the same as for the real quasi-momenta, it leads to Eqs. (169)–(173). Note that

$$c(k_{N+1}, L - y) = c^*(k_N, y), \tag{176}$$

and the involution (174) holds true.

We plot $\varrho(y)$ in Fig. 10. The top panels show that it oscillates if $Q \neq 0$. The bottom panels (d) and (e) demonstrate that the oscillations are largely, but not fully, suppressed for the function $e^{iQy}\varrho(y)$. Since the number of the gas particles, $N = 40$, used in the plot, is large, the residual oscillations seen in the bottom panels (d) and (e) can be attributed to the subleading term written explicitly on the right hand side of Eq. (119), valid in the thermodynamic limit. There are no visible oscillations in the bottom panel (f), consistent with the small contribution of the subleading terms to the asymptotic formula (128). Note that the oscillations of the function $e^{iQy}\varrho(y)$ can be seen in Fig. 4 from Ref. [23], though the thermodynamic limit have not been taken in the analytic formulas used therein, and the period of the oscillations has not been identified.

The transition from Eq. (173) to the Fredholm determinant representations (60) and (66) is straightforward, the details are given in appendix A.

# 9 Conclusion

The main result of the present paper is the Fredholm determinant representation, Eqs. (60) and (66), for the momentum distribution function, $n(k, Q)$, of an impurity which formed a polaron state with a free Fermi gas (or the Tonks-Girardeau gas [3,4]). Using this representation we examined how the properties of the impurity depend on the strength $g$ of the impurity-gas $\delta$-function interaction potential, and on the value of the total momentum $Q$ of the system

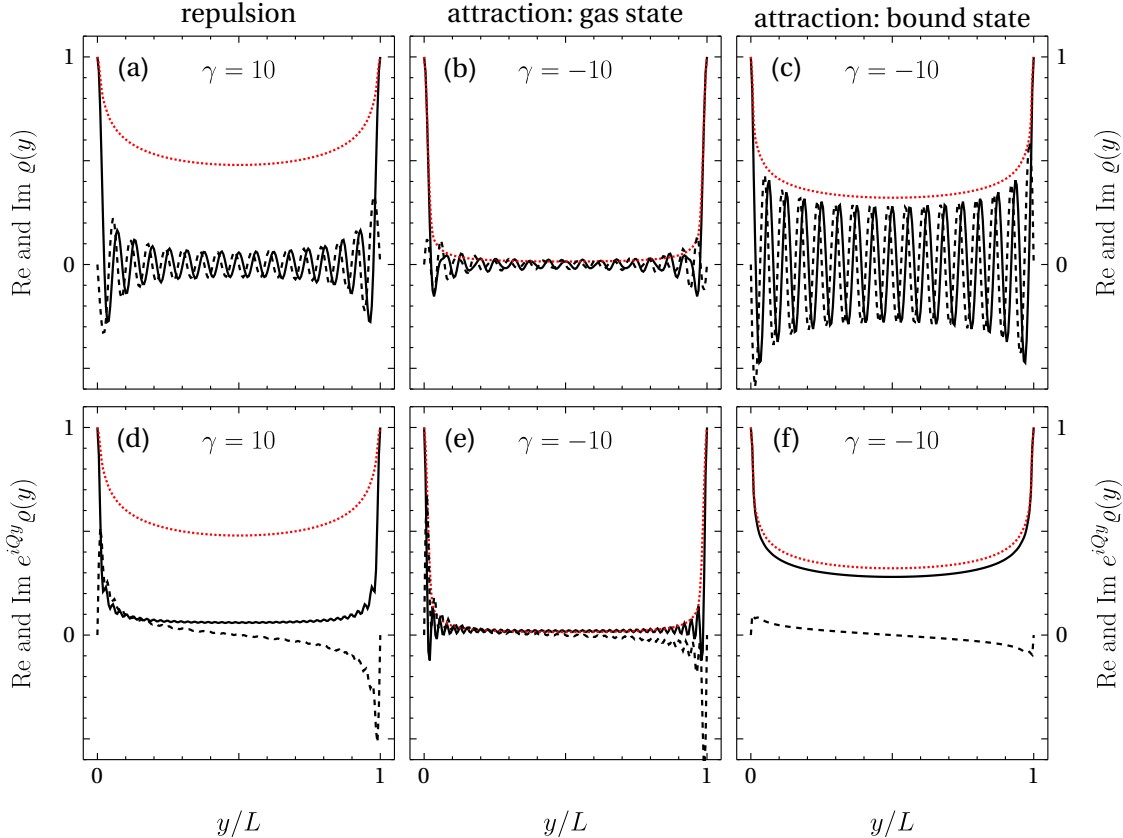

Figure 10: The reduced density matrix $\varrho(y)$ is examined for a gas with $N = 40$ particles. The red dotted lines are for $\varrho(y)$ at the total momentum $Q = 0$. The black solid (dashed) lines are for the real (imaginary) part of $\varrho(y)$ in the upper panels, and of $e^{iQy}\varrho(y)$ in the lower panels, at $Q = 0.8k_F$. Note how well the term $e^{iQy}$ suppresses the oscillations of $\varrho(y)$.

(which is the same as the momentum of the polaron). We have found that the formation of the bound state strongly affects the behavior of $n(k,Q)$. In the absence of the bound state $n(k,Q)$ turns into the Fermi function at $Q = 1$ (recall that the momenta are given in the units of the Fermi momentum $k_F$ everywhere except for the captions to the figures). This can be seen in Fig. 2(a) and 2(b). In the presence of the bound state $n(k,Q)$ has a weak singularity at $k = Q$ for any $Q$, including $Q = 1$, and the Fermi function does not emerge. This can be seen in Fig. 2(c). The distinct role the bound state plays in the behavior of the impurity's momentum distribution function is reflected at the level of the dispersion relation of the polaron. Indeed, the group velocity of the polaron vanishes at $Q = 1$ in the absence of the bound state, see Fig. (1)(a). Such a vanishing velocity is consistent with the impurity spreading over the Fermi sea and mimicking the distribution of the gas particles in the momentum space. In the presence of the bound state the group velocity of the polaron does not vanish at $Q = 1$, therefore the momentum distribution of the impurity cannot have the shape of the Fermi distribution function. Another distinct feature of the polaron in the presence of the bound state is almost linear dependence of the group velocity on $Q$ for all values of the coupling strength, Fig. 3(b). That is, the impurity can be viewed as a free particle having the effective mass $m_*$, and its momentum is $\langle P_{\rm imp}\rangle \simeq Q/m_*$. This is also seen from perturbative calculations, Ref. [49], not limited to the exactly solvable case considered in our paper.

We have used the exact wave functions and spectrum of the model. In a number of papers the mobile impurity problem is investigated by using approximate wave functions. Being con-

structed from a few particle-hole excitations, Refs. [50, 51], these functions predict rather accurately some static properties, Ref. [52], and time dynamics, Fig S4 in Ref. [37], of the mobile impurity in one dimension. The momentum distribution function has not been treated using the aforementioned basis of the variation functions, to the best of our knowledge. Other natural ways to construct variation functions, by taking solely a product of coherent states [53–55], including Gaussian state correlations between different momentum modes [56], or correlations to an arbitrarily high order [57], are also promising. How to perform a resummation of the excitations containing arbitrary number of the particle-hole pairs for a weak impurity-gas coupling is discussed in Refs. [58–60].

An exciting development of ultracold atomic physics made it possible to setup experiments on diffusion and drag of quantum impurities embedded in a degenerate ultracold gas. Special to one dimension is the observation of the Bloch oscillations of a mobile impurity moving through a quantum fluid in the absence of a periodic lattice [61]. The momentum distribution function of the impurity has been measured in that experiment. However, there the impurity neither started out in the equilibrium ground state, nor reached such a state in the course of the temporal evolution.

# Acknowledgements

We thank Vadim Cheianov, Eugene Demler, Pavel Dolgirev, Eoin Quinn, Michael Knap, and Ovidiu Pâţu for their valuable comments to this work. We also acknowledge the comments from the referees, which helped us to improve the manuscript.

**Funding information** O.L. acknowledges the support from the Russian Foundation for Basic Research under the grant N 18-32-20218. The work of M. B. Z. is supported by Grant No. ANR-16-CE91-0009-01 and CNRS grant PICS06738.

# A  The $L \to \infty$ limit of Eq. (173): transformation to Eqs. (60) and (66)

In this appendix we explain how we arrive at the Fredholm determinant representation (60) and (66), valid for $N \to \infty$, starting from Eq. (173), valid for any finite $N$. Recall that we are working at a finite gas density, therefore $N \to \infty$ implies $L \to \infty$.

Combining the definitions (30) and (65) we write

$$e^{i\delta(q)} = -\left[\frac{\nu_+(q)}{\nu_-(q)}\right]^{1/2}, \qquad \sin\delta(q) = [\nu_+(q)\nu_-(q)]^{1/2}. \qquad (177)$$

The $L \to \infty$ limit of Eq. (32) reads

$$\frac{\partial k_j}{\partial \Lambda} = \frac{2}{L}\nu_-(k_j)\nu_+(k_j), \qquad j = 1, \dots, N+1, \qquad L \to \infty \qquad (178)$$

for the real $k_1, \dots, k_{N+1}$. This way, we get the kernel (61) from Eq. (169). Combining Eqs. (152) and (44) we have

$$\sum_{j=1}^{N+1} \frac{\partial k_j}{\partial \Lambda} = Z. \qquad (179)$$

This way, we get the kernel (62) from Eq. (172). This completes the derivation of the Fredholm determinant representation (60).

Now let us turn to the derivation of Eq. (66). The quasi-momenta $k_N$ and $k_{N+1}$ are now complex, $k_N^* = k_{N+1}$. Combining Eqs. (152) and (51) we have

$$\sum_{j=1}^{N+1} \frac{\partial k_j}{\partial \Lambda} = Z_b. \tag{180}$$

Recall that $Z_b < 0$. The $L \to \infty$ limit of $k_N$ and $k_{N+1}$ is given by Eq. (49). The leading term in the large $L$ expansion of Eq. (160) in the interval $0 \leq y \leq L$ is

$$c(k_N) \to c(k_+) = e(k_+) = 2i e^{ik_+(y-L)}, \qquad L \to \infty, \tag{181}$$

and

$$c(k_{N+1}) \to c(k_-) = e(k_-) = -2i e^{ik_- y}, \qquad L \to \infty. \tag{182}$$

Substituting equation (49) into (32) we obtain

$$\frac{\partial k_N}{\partial \Lambda} = \frac{\partial k_{N+1}}{\partial \Lambda} = \frac{1}{\alpha} + \mathcal{O}(e^{-|g|L}) \tag{183}$$

in place of Eq. (178) for $j = N, N+1$. Further, we limit $y$ to the interval $0 \leq y \leq L/2$, which implies $e(k_+) = 0$ for Eq. (181). This gives

$$e_+(k_N) \to e_+(k_+) = 0, \qquad e_+(k_{N+1}) \to e_+(k_-) = \frac{2i}{\pi} e^{ik_- y/2} \left| \frac{L}{2\alpha} \right|^{1/2}, \tag{184}$$

and

$$e_-(k_N) \to e_-(k_+) = e^{-ik_+ y/2} \left| \frac{L}{2\alpha} \right|^{1/2}, \qquad e_-(k_{N+1}) \to e_-(k_-) = e^{-ik_- y/2} \left| \frac{L}{2\alpha} \right|^{1/2} \tag{185}$$

for the $L \to \infty$ limit of the functions $e_\pm$ defined by Eq. (170). Evidently,

$$e_+(k_j) = -\frac{1}{\pi} \frac{e^{ik_j y}}{v_-(k_j)} e_-(k_j), \qquad e_+(k_-) = \frac{2i}{\pi} e^{ik_- y} e_-(k_-). \tag{186}$$

Therefore, we get for the $L \to \infty$ limit of the function (169)

$$K(k_j, k_N) = -\frac{1}{\pi} \frac{e^{ik_j y}}{v_-(k_j)} \frac{e_-(k_j) e_-(k_+)}{k_j - k_+}, \qquad j = 1, \ldots, N-1, \tag{187}$$

and

$$K(k_j, k_{N+1}) = -\frac{1}{\pi} \left[ \frac{e^{ik_j y}}{v_-(k_j)} + 2i e^{ik_- y} \right] \frac{e_-(k_j) e_-(k_-)}{k_j - k_-}, \qquad j = 1, \ldots, N-1, \tag{188}$$

and

$$K(k_N, k_{N+1}) = -\frac{\alpha}{\pi} e^{ik_- y} e_-(k_+) e_-(k_-). \tag{189}$$

For the diagonal terms we use Eq. (165)

$$A_{NN} = 0, \qquad A_{N+1N+1} = \frac{2y}{\alpha} \tag{190}$$

and combine it with Eq. (168). This gives

$$K(k_N, k_N) = \frac{\alpha}{\pi} e^{ik_+ y} [e_-(k_+)]^2, \qquad K(k_{N+1}, k_{N+1}) = \frac{\alpha}{\pi} e^{ik_- y} [e_-(k_-)]^2 \left( 1 - \frac{2y}{\alpha} \right). \tag{191}$$

Using the identity

$$\det(M + \xi R) = (1 - \xi)\det M + \xi \det(M + R), \tag{192}$$

where $\xi$ is a number, and $R$ is a rank one matrix, we write Eq. (173) as

$$\varrho(y) = \partial_\xi \det\left(I + \frac{2\pi}{L}K + \xi\frac{2\pi}{L}W\right)\Big|_{\xi=0}. \tag{193}$$

The two last rows and columns of this matrix are special because $k_N$ and $k_{N+1}$ are complex:

$$\det\left(I + \frac{2\pi}{L}K + \xi\frac{2\pi}{L}W\right) = [e_-(k_-)e_-(k_+)]^2 \det\left(\begin{array}{c|c|c} \mathcal{A} & \mathcal{A}^+ & \mathcal{A}^- \\ \hline \mathcal{A}^+ & a & b \\ \hline \mathcal{A}^- & b & d \end{array}\right). \tag{194}$$

Here,

$$\mathcal{A}_{jl} = \delta_{jl} + \frac{2\pi}{L}K_{jl} + \frac{2\pi}{L}\frac{\xi}{Z_b}\frac{e_-(k_j)e_-(k_l)}{\pi}, \qquad j, l = 1, \ldots, N-1, \tag{195}$$

$$\mathcal{A}_j^+ = \frac{2\pi}{L}\frac{e_-(k_j)}{\pi}\left(-\alpha e^{ik_j y} + \frac{\xi}{Z_b}\right), \qquad j = 1, \ldots, N-1, \tag{196}$$

$$\mathcal{A}_j^- = \frac{2\pi}{L}\frac{e_-(k_j)}{\pi}\left[-f_1(k_j) + \frac{\xi}{Z_b}\right], \qquad j = 1, \ldots, N-1, \tag{197}$$

and

$$\mathcal{D} \equiv \begin{pmatrix} a & b \\ b & d \end{pmatrix} = -\frac{2\pi}{L}\frac{\alpha}{\pi}e^{ik_- y}\begin{pmatrix} 0 & 1 \\ 1 & \frac{2y}{\alpha} \end{pmatrix} + \frac{2\pi}{L}\frac{\xi}{\pi Z_b}\begin{pmatrix} 1 & 1 \\ 1 & 1 \end{pmatrix}, \tag{198}$$

where

$$f_1(q) = \frac{e(q) - e(k_-)}{k_j - k_-}. \tag{199}$$

We calculate the determinant and the inverse of $\mathcal{D}$ omitting the terms which are higher than the first order in $\xi$:

$$\det\mathcal{D} = \left(\frac{2\pi}{L}\right)^2 e^{ik_- y}\frac{\alpha^2}{\pi^2}\left[-e^{ik_- y} + \frac{\xi}{Z_b}\frac{2}{\alpha}\left(1 - \frac{y}{\alpha}\right)\right] \tag{200}$$

and

$$\mathcal{D}^{-1} = \frac{L}{2\pi}\frac{\pi}{\alpha}e^{-ik_- y}\left[\begin{pmatrix} \frac{2y}{\alpha} & -1 \\ -1 & 0 \end{pmatrix} - \frac{\xi}{Z_b}\frac{e^{-ik_- y}}{\alpha}\begin{pmatrix} \left(1 - \frac{2y}{\alpha}\right)^2 & 1 - \frac{2y}{\alpha} \\ 1 - \frac{2y}{\alpha} & 1 \end{pmatrix}\right]. \tag{201}$$

Suppose $\mathcal{A}, \mathcal{B}, \mathcal{C}$, and $\mathcal{D}$ are arbitrary matrices of dimension $n \times n$, $n \times m$, $m \times n$, and $m \times m$, respectively. When $\mathcal{D}$ is invertible, one has the identity

$$\det\begin{pmatrix} \mathcal{A} & \mathcal{B} \\ \mathcal{C} & \mathcal{D} \end{pmatrix} = \det(\mathcal{D})\det(\mathcal{A} - \mathcal{B}\mathcal{D}^{-1}\mathcal{C}). \tag{202}$$

We use this identity for the determinant (194), where $\mathcal{D}$ is given by Eq. (198). We have

$$[\mathcal{B}\mathcal{D}^{-1}\mathcal{C}]_{jl} = \mathcal{A}_j^+\mathcal{D}_{11}^{-1}\mathcal{A}_l^+ + \mathcal{A}_j^-\mathcal{D}_{21}^{-1}\mathcal{A}_l^+ + \mathcal{A}_j^+\mathcal{D}_{12}^{-1}\mathcal{A}_l^- + \mathcal{A}_j^-\mathcal{D}_{22}^{-1}\mathcal{A}_l^-. \tag{203}$$

This gives

$$[\mathcal{B}\mathcal{D}^{-1}\mathcal{C}]_{jl} = \frac{2\pi}{L} \frac{e_-(k_j)e_-(k_l)}{\pi} e^{-ik_-y} \Big\{ -f_1(k_j)e^{ik_ly} - e^{ik_jy}f_1(k_l) + 2y e^{ik_jy}e^{ik_ly}$$
$$-\frac{\xi}{Z_b}\frac{e^{-ik_-y}}{\alpha^2}[f_2(k_j)f_2(k_l) - \alpha^2 e^{2ik_-y}] \Big\}, \quad (204)$$

where

$$f_2(q) = \alpha e^{ik_-y} - (\alpha - 2y)e^{iqy} - f_1(q). \quad (205)$$

This leads to the following representation of Eq. (193) in the $L \to \infty$ limit:

$$\varrho(y) = e^{-ik_+y} \, \partial_\xi \left\{ \left[ -e^{ik_-y} + \frac{\xi}{Z_b}\frac{2}{\alpha}\left(1 - \frac{y}{\alpha}\right) \right] \det\left(\hat{I} + \hat{K}_1 + \xi\hat{V}_1\right) \right\}\bigg|_{\xi=0}. \quad (206)$$

Here,

$$K_1(q,q') = K(q,q') + \frac{e_-(q)e_-(q')}{\pi}e^{-ik_-y}\left[ f_1(q)e^{iq'y} + f_1(q')e^{iqy} - 2y e^{i(q+q')y} \right], \quad (207)$$

and

$$V_1(q,q') = \frac{e^{-2ik_-y}}{\pi\alpha^2 Z_b}e_-(q)e_-(q')f_2(q)f_2(q'). \quad (208)$$

Using the identity (192) we transform Eq. (206) to

$$\varrho(y) = e^{-i(k_++k_-)y}\left[ \det(\hat{I} + \hat{K}_1 + \hat{W}_1) - c\det(\hat{I} + \hat{K}_1) \right]$$
$$= e^{-i(k_++k_-)y}(1-c)\det\left( \hat{I} + \hat{K}_1 + \frac{1}{1-c}\hat{W}_1 \right), \quad (209)$$

where

$$W_1(q,q') = -\frac{1}{\pi\alpha^2 Z_b}e_-(q)e_-(q')f_2(q)f_2(q'), \quad (210)$$

and

$$c = 1 - \frac{2e^{ik_-y}(\alpha - y)}{\alpha^2 Z_b}. \quad (211)$$

One has

$$K_1(q,q') + \frac{1}{1-c}W_1(q,q') = K_b(q,q') + \frac{1}{1-c}W_b(q,q'), \quad (212)$$

where

$$K_b(q,q') = K(q,q') + \frac{\alpha}{\pi}e_-(q)e_-(q')(e^{iqy} + e^{iq'y}), \quad (213)$$

$$W_b(q,q') = -\frac{e_-(q)e_-(q')f(q)f(q')}{\pi\alpha^2 Z_b}, \quad (214)$$

and

$$f(q) = \alpha(e^{ik_-y} + e^{iqy}) - f_1(q). \quad (215)$$

We get for Eq. (209)

$$\varrho(y) = e^{-i(k_++k_-)y}(1-c)\det\left( \hat{I} + \hat{K}_b + \frac{1}{1-c}\hat{W}_b \right). \quad (216)$$

Using the identity (192) we arrive at the expression

$$\varrho(y) = e^{-i(k_++k_-)y}\left[ \det\left(\hat{I} + \hat{K}_b + \hat{W}_b\right) - c\det\left(\hat{I} + \hat{K}_b\right) \right]. \quad (217)$$

This is the desired representation (66).

# B  Small distance expansion of the reduced density matrix

In this appendix we derive formulas presented in section 6. We start from the finite-size expression for the reduced density matrix, Eq. (173). This way the repulsive ground state, the attractive gas state, and the attractive bound state are treated all at once.

The expansion of the kernels (169) and (172) up to order three in $y$ reads

$$
\frac{2\pi}{L} \left| \frac{\partial k_j}{\partial \Lambda} \frac{\partial k_l}{\partial \Lambda} \right|^{-1/2} K(k_j, k_l) = -\alpha + i(i+\Lambda)y - \frac{i}{2} y\alpha(k_j + k_l)
$$
$$
+ \left[ \frac{\alpha}{8} y^2 - \frac{i}{24}(i+\Lambda)y^3 \right](k_j^2 - 2k_j k_l + k_l^2) + \frac{i}{48}\alpha y^3 (k_j^3 - k_j^2 k_l - k_j k_l^2 + k_l^3) + \cdots, \quad (218)
$$

and

$$
\frac{2\pi}{L} \left( \sum_{m=1}^{N+1} \frac{\partial k_m}{\partial \Lambda} \right) \left| \frac{\partial k_j}{\partial \Lambda} \frac{\partial k_l}{\partial \Lambda} \right|^{-1/2} W(k_j, k_l) = 1 - \frac{i}{2} y(k_j + k_l)
$$
$$
- \frac{1}{8} y^2 (k_j^2 + 2k_j k_l + k_l^2) + \frac{i}{48} y^3 (k_j^3 + 3k_j^2 k_l + 3k_j k_l^2 + k_l^3) + \cdots, \quad (219)
$$

respectively.

After substituting the expansions (218) and (219) into the determinants on the right hand side of Eq. (173) we use the following identity

$$
\det_N(I + UV^T) = \det_s(I + V^T U). \quad (220)
$$

Here, $U$ and $V$ are $N \times s$ matrices with the columns formed by $N+1$-component vectors $u_1, \ldots, u_s$ and $v_1, \ldots v_s$, respectively. As a result (MATHEMATICA package has been used to evaluate the determinants) we expanded Eq. (173) up to order three in $y$:

$$
\varrho(y) = 1 + \frac{-iy}{S_0} S_1 + \frac{(-iy)^2}{2S_0} \left( S_2 - \alpha S_0 S_2 + \alpha S_1^2 \right)
$$
$$
+ \frac{(-iy)^3}{6S_0} \left[ S_3 - (\Lambda + i)(S_0 S_2 - S_1^2) - \alpha S_0 S_3 + \alpha S_1 S_2 \right] + \cdots, \quad (221)
$$

where

$$
S_n = \sum_{j=1}^{N+1} k_j^n \frac{\partial k_j}{\partial \Lambda}. \quad (222)
$$

We now take the thermodynamic limit in Eq. (222). For the repulsive ground state and the attractive gas state we have

$$
S_0 = Z, \quad (223)
$$

$$
S_1 = \frac{\Lambda}{\alpha} Z + \varphi, \quad (224)
$$

$$
S_2 = \frac{\Lambda^2 - 1}{\alpha^2} Z + \frac{2}{\pi \alpha^2} + \frac{2\Lambda}{\alpha} \varphi, \quad (225)
$$

$$
S_3 = \frac{\Lambda^3 - 3}{\alpha^3} \Lambda Z + \frac{4\Lambda}{\pi \alpha^3} + \frac{3\Lambda^2 - 1}{\alpha^2} \varphi. \quad (226)
$$

Here,

$$
\varphi = \frac{1}{2\pi \alpha^2} \ln \frac{1 + (\alpha - \Lambda)^2}{1 + (\alpha + \Lambda)^2}, \quad (227)
$$

and $Z$ is given by Eq. (64). Notably, the result for the attractive bound state follows by just replacing $Z$ with $Z_b$, Eq. (71).

Finally, the expansion (221) gives for Eqs. (98), (99), and (102)

$$\langle P_{\text{imp}} \rangle = \frac{S_1}{S_0}, \tag{228}$$

$$\langle P_{\text{imp}}^2 \rangle = \frac{S_2 + \alpha(S_1^2 - S_0 S_2)}{S_0}, \tag{229}$$

and

$$C = \frac{S_2 S_0 - S_1^2}{\pi S_0}, \tag{230}$$

respectively. This leads us to the results discussed in section 6.

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
