# Peer review of "Zero temperature momentum distribution of an impurity in a polaron state of one-dimensional Fermi and Tonks-Girardeau gases"

_SciPost Physics, doi:SciPost Phys. 8, 053 (2020)_

## Round 2 · Referee Report · Anonymous (Referee 1) · 2019-12-6

Strengths

1.Clear presentation of technically involved results.
2.Relevant for both theoreticians and experimentalists working on ultracold gases.

Report

This is a clear and well written paper in which the authors derive a Fredholm determinant
representation for the momentum distribution of a impurity interacting with a free Fermi
gas in one-dimension. The impurity interacts with the free gas via a delta-function potential
(repulsion or attraction of arbitrary strength g) and the authors consider the polaron state
(minimum energy state for a given total momentum Q) at zero temperature.
From the determinant representation they derive:
a) The average momentum, root mean uncertainty and the $C/k^4$ tails of the momentum distribution.
b) The singularity of n(k,Q) at k=Q.
c) Establish the correspondence of n(k,Q) and the field-field correlation function of 1D
impenetrable anyons in the $g\rightarrow \infty$ limit.
The paper reports relevant, original and comprehensive results particularly impressive being the
determination of the $\nu$ exponent. I recommend the publication of this article in SciPost Physics.

Requested changes

1.The authors state in the title and abstract that their results are valid for an impurity
immersed in a free Fermi or Tonks-Girardeau gas. The results presented in the paper are
derived for the case of the free Fermi gas. While intuitively the same results should hold
for the case when the gas is formed by Tonks-Girardeau bosons the paper should contain at
least a reference on why this is the case (like Section 2 of Ref. 25).

2.Typos:

page 3 line 4: free Fremi -> free Fermi
page 17 after Eq. 108: contrased -> contrasted
page 24 Title of Sect. 7.4 exponenent -> exponent

---

## Round 2 · Referee Report · Anonymous (Referee 2) · 2019-12-11

Strengths

  1. New and interesting results.
  2. Interesting perspectives (edge behaviour from the form factor analysis)

Weaknesses

  1. Organisation of the paper
  2. Very particular case where the result can be applied (one impurity particle sector).

Report

The paper treats the correlation functions of the two-component Fermi-gas with delta interaction in a special case with only one particle of second type, considered is an impurity. The authors obtain a determinant representation for the two-point equal-time correlation function for the impurity particle and the momenta distribution as the Fourier transform of this correlation function. Then the authors check the result computing various limiting cases where the computation can be computed using alternative techniques. Finally they announce the most interesting result of the paper on the edge behaviour of the momentum distribution and large distance behaviour of the correlation function. These last results are not obtained from the determinant representation but from the form factor analysis (and the derivation is not given in the current paper), however it seems to be reasonable to present them in this article.

The results of the paper are new and interesting, personally I have a problem with the order of presentation (main results - limiting cases - edge behaviour - proof of the main result), but otherwise the paper is easily readable. In conclusion I recommend the paper for publication.

Requested changes

  1. I suggest (it is optional but I think it can considerably improve the readability of the paper) to change the presentation order. It would be much more logical to give the derivation of the main result (section 8) directly after the section presenting it and only then consider the limiting cases (sections 4-6). It seems also logical to give the announcement of the asymptotic result (section 7) in the last section as it is supposed to be a link to a future publication. It will also provide a possibility to refer to the form factor expansion used in the derivation of the main result as the starting point of the asymptotic analysis.

  2. I would also suggest to give more details in the derivation of the main result. In particular the part between eqs. (157) and (159) as it is the least straightforward part of the derivation. For example in the equation (158) different arguments should be used to show that 2 integrals are zero in the limit and this discussion is completely skipped by the authors. It would be useful also to mention the order of corrections for both integrals.

---

## Round 2 · Referee Report · Anonymous (Referee 3) · 2019-12-28

Strengths

Timely topic
Very well presented
Impressive calculations

Report

In their manuscript, Gamayun et al. have analytically calculated the momentum distribution function of a single mobile impurity interacting with a surrounding gas of free fermions. This problem is exactly solvable by the Bethe ansatz, and the authors took the non-trivial challenge to calculated the full momentum (k) distribution function n(k,Q), where Q is the total system, or polaron, momentum. This paper deserves to be published in SciPost Physics. The calculations are explained at great detail, and the results are very clearly presented. (For fairness, I should mention that — while I’m not a complete novice to the Bethe ansatz — I am not an expert either. So I couldn’t check these impressive calculations in any detail. But knowing some of the author’s other publications, I fully trust their results.) Most of all, the topic of the paper is timely: Using ultracold atoms, distribution functions of the type discussed by the authors could be relatively easily measured, for example. As discussed in the end of the manuscript by the authors, other — e.g. semi-analytical — methods could be used, too, to calculate similar distribution functions. This has not been done before, so the present paper is really pioneering this field. More importantly, the calculations performed here provide a most valuable benchmark for future studies. While extensions to more general cases (e.g. with unequal impurity masses, or considering bosonic background gases with mutual interactions, etc.) may not be addressable by the Bethe ansatz solution anymore, or only partly, other methods might be able to do so. However, to benchmark and develop such methods, the results obtained in this present manuscript are invaluable. I fully recommend publication of this manuscript in its present form.

The only suggestion I have is that the authors could add a brief discussion how the distribution function n(k,Q) could be measured experimentally. Ultracold atoms provide one promising avenue, but might not be the only option.

In Eq. (25), j+1 should probably be replaced by j=1 in the sum.

Requested changes

None.

---

## Round 3 · Referee Report · Anonymous (Referee 1) · 2020-3-10

Report

After the revision, which incorporated the suggestions of all three referees,
I feel that the paper has improved significantly and, therefore, I warmly recommend its publication in SciPost Physics.

---

## Round 3 · Referee Report · Anonymous (Referee 3) · 2020-3-10

Report

I recommend that the revised version of this manuscript should be published in its present form.

---

## Round 3 · Referee Report · Anonymous (Referee 2) · 2020-3-17

Strengths

-

Weaknesses

-

Report

The authors improved the presentation of the section 8. As the suggestion to change the order of sections was optional, I recommend this new version for publication

---

## Round 3 · Author Response

Response to Referee 1:

  1. Indeed, apart from the title and the abstract, we have mentioned the Tonks-Girardeau bosons only in the conclusions, without much explanation. Now, we are mentioning the Tonks-Girardeau bosons in the beginning of section 2, and give a reference to the section 2 of the work [25].

  2. Typos corrected

Response to Referee 2:

  1. We understand the logic of the proposed change. However, our point is that sections 3 to 7 are a natural continuation of the section 2 in that they altogether contain the result: the analytic formulas used to describe the momentum distribution function of the impurity. Hence, the reader who only wants the result about the momentum distribution may omit reading the last section 8. The above said, we would like to keep the order of the sections of the manuscript unchanged.

  2. We have explained vanishing of the integrals in Eq. (158) in more detail. Furthermore, section 8 contains now only formulas valid for any N. All calculations relevant for the large N limit are put into the appendix A. The title of this appendix has been changed accordingly.

Response to Referee 3:

We have added a paragraph to the "Conclusion" section discussing in brief the experimental findings in ultracold atomic gases related to the observable studied in our manuscript.

---

## Round 3 · List of Changes

Restored overall phase factor in Eq. (66), accidentally erased in the course of the editing of the manuscript. This does not affect any other equation or figure in the manuscript. Some notations are changed around Eqs. (66)-(71), and through the appendix A. Title change for the appendix A.

References [36] and [61] added

Beginning of the section 2 contains now an explanation of why our results are valid for the Tonks-Girardeau gas

Conclusion section contains now a paragraph discussing an experiment in ultracold atomic gases

An acknowledgement to Referees is added to the Acknowledgements section

Various minor stylistic/language changes, typos corrected

---

## Editorial Decision

published